# Empirical Study on Robustness and Resilience in Cooperative Multi-Agent Reinforcement Learning

**Simin Li[1], Zihao Mao[1], Hanxiao Li[1], Zonglei Jing[1], Zhuohang Bian[1], Jun Guo[1], Li Wang[1]**
**Zhuoran Han[1], Ruixiao Xu[1], Xin Yu[1], Chengdong Ma[3], Yuqing Ma[5], Bo An[6]**
**Yaodong Yang[3] \*, Weifeng Lv[1] \*, Xianglong Liu[1, 2, 4] \***

[1]State Key Laboratory of Complex & Critical Software Environment, Beihang University, China
[2]Zhongguancun Laboratory, China   [3]Institute of Artificial Intelligence, Peking University, China
[4]Institute of Data Space, Hefei Comprehensive National Science Center, China
[5]Institute of Artificial Intelligence, Beihang University, China
[6]Nanyang Technological University, Singapore

## Abstract

In cooperative Multi-Agent Reinforcement Learning (MARL), it is a common practice to tune hyperparameters in ideal simulated environments to maximize cooperative performance. However, policies tuned for cooperation often fail to maintain robustness and resilience under real-world uncertainties. Building trustworthy MARL systems requires a deep understanding of *robustness*, which ensures stability under uncertainties, and *resilience*, the ability to recover from disruptions—a concept extensively studied in control systems but largely overlooked in MARL. In this paper, we present a large-scale empirical study comprising over 82,620 experiments to evaluate cooperation, robustness, and resilience in MARL across 4 real-world environments, 13 uncertainty types, and 15 hyperparameters. Our key findings are: (1) Under mild uncertainty, optimizing cooperation improves robustness and resilience, but this link weakens as perturbations intensify. Robustness and resilience also varies by algorithm and uncertainty type. (2) Robustness and resilience do not generalize across uncertainty modalities or agent scopes: policies robust to action noise for all agents may fail under observation noise on a single agent. (3) Hyperparameter tuning is critical for trustworthy MARL: surprisingly, standard practices like parameter sharing, GAE, and PopArt can hurt robustness, while early stopping, high critic learning rates, and Leaky ReLU consistently help. By optimizing hyperparameters only, we observe substantial improvement in cooperation, robustness and resilience across all MARL backbones, with the phenomenon also generalizing to robust MARL methods across these backbones. Code and results available at `https://github.com/BUAA-TrustworthyMARL/adv_marl_benchmark`.

## 1   Introduction

Cooperative Multi-agent reinforcement learning (MARL) [1, 2, 3, 4, 5, 6, 7, 8, 9, 10] has demonstrated significant success across various complex simulation environments, including StarCraft [11], turbulent flow modeling [12], and network control [13]. However, real-world deployment of MARL systems introduce uncertainties unseen in simulation environments. These uncertainties spans from observation errors [14, 15], action perturbations [16, 17], and environmental unpredictability [18, 19], which degrade the performance of MARL significantly.

In this paper, we study two fundamental ways to deal with uncertainties in real-world deployment: *robustness* and *resilience*. Robustness measures the ability of MARL to withstand uncertainties in the

---

\*Corresponding Authors. E-mails: yaodong.yang@pku.edu.cn, lwf@buaa.edu.cn, xlliu@buaa.edu.cn.

39th Conference on Neural Information Processing Systems (NeurIPS 2025).

real-world, enabling MARL systems to maintain functionality despite partial system failures [20]. Resilience, on the other hand, is the ability of MARL to recover from disruptions. For example, robot swarms may lose coordination due to communication interference, or smart grids may collapse during natural disasters. After the disruption, the system should be able to recover from such external shock. While robustness and resilience are complementary, their distinction has been extensively studied in fields such as control theory [21], ecology [22], and economics [23], but remains underexplored in MARL. Current MARL research often conflates these concepts [24, 25, 26], neglecting their critical differences and failing to account for resilience in evaluation.

Another challenge in dealing with real-world uncertainties lies in understanding the role of hyperparameters. While often overlooked in theoretical analyses, careful selection of hyperparameters can play a more critical role than the algorithms themselves. In single-agent RL, hyperparameters account for most of the observed empirical differences between TRPO and PPO [27]. Similarly, in MARL, MAPPO achieves state-of-the-art performance largely due to its emphasis on hyperparameters [3], including parameter sharing and five PPO-specific choices. Consequently, it is common practice for MARL researchers to tune hyperparameters to optimize cooperative performance.

However, such tuning is effective only in the idealized environments on which it is trained, and its effects on policy robustness and resilience remain largely unexplored. As shown in Fig. 1, while changing hyperparameters maximizes cooperative rewards, it significantly reduces robustness and resilience under uncertainties. Further analysis shows that hyperparameters influence cooperation, robustness and resilience more strongly than the choice of MARL algorithm in 9 out of 18 tasks via two-way ANOVA ($p < 0.001$; see Appendix. B.4 for full results). To evaluate and understand the role of robustness and resilience in MARL, we conduct a large-scale empirical study involving 82,620 experiments across 4 real-world environments, 13 uncertainty types, and 15 hyperparameters. Our main findings are as follows:

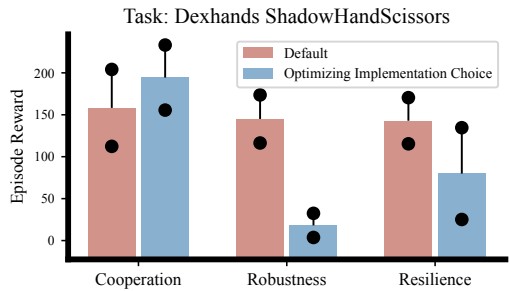

Figure 1: While optimizing hyperparameters to improve cooperation, the algorithm gets significantly less robust and resilient when uncertainty occurs.

1. While improving cooperation can enhance robustness and resilience under mild uncertainty, this relationship weakens as perturbations intensify. This trend holds across diverse uncertainty types and agent configurations. Moreover, algorithm sensitivity varies: MADDPG is more robust to action noise, whereas MAPPO and HAPPO perform better under observation uncertainty.

2. Robustness in MARL can not generalize across uncertainty types or scopes. Policies trained to handle action noise may still fail under observation and environment shifts, and defenses against group-level perturbations may break down under attacks against single agent. Trustworthy MARL must account for diverse uncertainty modalities and agent-level effects.

3. Hyperparameter tuning plays a critical role in robustness and resilience. Surprisingly, common practices such as parameter sharing, GAE, and PopArt can hurt performance under uncertainty, while techniques like early stopping, critic-dominant learning rates, and Leaky ReLU consistently improve it. By combining these hyperparameters, we observe an average improvement of 52.60% in cooperation, 34.78% in robustness and 60.34% in resilience. This same set of hyperparameters generalizes to robust MARL methods on the same backbones, yielding average improvements of 89.43% on cooperation, 65.83% on robustness, 82.96% on resilience.

## 2 Related Work

**Uncertainties in RL and MARL** Robustness against uncertainties has been extensively studied within the framework of robust RL and MARL, as summarized in [28]. In reinforcement learning, uncertainties in deployment are typically categorized into state [29, 30, 31], action [32], and environment uncertainties [33, 18]. MARL follows a similar taxonomy but incorporates the multi-agent dynamic. For observation uncertainties, noise can be introduced to each agent [34, 15, 35] or applied to specific agents [14]. Action uncertainties often arise when one or more agents deviate from their

optimal policy [16, 36, 37, 17]. Environment uncertainties in MARL are typically analogous to those seen in single-agent RL [38, 19]. The primary focus of robust RL and MARL is to make algorithms robust against the worst-case adversaries chosen in the uncertainty set. In our paper, we select a set of representative methods derived from these studies for robustness evaluation. For resilience, the MARL literature often overlook its distinct role from robustness, using the term resilience and robustness interchangeably [25, 39, 26, 40]. In this paper, we draw in fields such as control theory [21], ecology [22], and economics [23] to formally define resilience in MARL.

**The Importance of hyperparameters.** It is a well-known fact that implementation matters for RL and MARL. In RL, [27] highlighted the surprising finding that variations in performance among RL algorithms often stem from hyperparameters rather than fundamental algorithmic differences. This notion was further explored by [41], who provided a comprehensive set of recommendations for RL implementation through large-scale experimentation. In MARL, Epymarl [42] introduced the first extensive benchmark suite, while MAPPO [3] significantly boosted Epymarl's performance by simply optimizing the implementation. Similarly, Pymarlv2 [43] demonstrated that state-of-the-art results in the SMAC environment can be achieved by fine-tuning QMIX [2]. [24] and [44] offer preliminary evaluations of RL/MARL under uncertainty, but are limited to simple simulation experiments in a small scale. The works most similar to ours are RRLS [45] and Robust Gymnasium [46], which provide integrated codebases for evaluating the robustness of single- and multi-agent RL. Our work differs by presenting the relations between cooperation, robustness and resilience under multiple real world environments, algorithms, and diverse uncertainties based on over 82,620 experiments.

## 3 Robustness and Resilience

**Preliminaries.** We formulate MARL as a decentralized partially observable Markov decision process (Dec-POMDP) [47], defined as a tuple:

$$\mathcal{G} = \langle \mathcal{N}, \mathcal{S}, \mathcal{O}, O, \mathcal{A}, \mathcal{P}, R, \gamma \rangle. \tag{1}$$

Here $\mathcal{N} = \{1, ..., N\}$ is the set of $N$ agents, $\mathcal{S}$ is the global state space, $\mathcal{O} = \times_{i \in \mathcal{N}} \mathcal{O}^i$ is the observation space, $O$ is the observation emission function. At each timestep, each agent selects an action $a^i \in \mathcal{A}^i$, with $\mathcal{A} = \times_{i \in \mathcal{N}} \mathcal{A}^i$ is the joint action space, forming a joint action $\mathbf{a} \in \mathcal{A}$. The environment proceeds following the state transition probability $\mathcal{P} : \mathcal{S} \times \mathcal{A} \to \Delta(\mathcal{S})$, and each agent receives a shared reward function $r(s, \mathbf{a}) : \mathcal{S} \times \mathcal{A} \to \mathbb{R}$ in the training environment, $\gamma \in [0, 1)$ is the discount factor. Each agent owns a state-action trajectory $\tau^i \in T \equiv (\mathcal{O}^i \times \mathcal{A}^i)^*$, and make decisions using a (stochastic) policy $\pi^i(a^i | \tau^i) : T \to \Delta(\mathcal{A}^i)$. Let $\rho_0$ be the initial state distribution, the goal of the joint policy $\pi$ is to maximize the objective $J(\pi) = \mathbb{E}_{s_0 \sim \rho_0}[\mathbb{E}_\pi[\sum_{t=0}^\infty \gamma^t r_t | s_0]]$.

**A Motivating Example: Robustness versus Resilience.** We use a metropolitan power grid to illustrate robustness and resilience under both small and large uncertainties. Under normal conditions, the grid faces minor perturbations such as sensor noise, voltage fluctuations, or local demand changes, where it should exhibit *robustness*, maintaining stable operation despite small disturbances. Yet even mild perturbations can temporarily degrade performance, requiring *resilience* to recover normal functionality. In contrast, rare but severe events like earthquakes or major short circuits may cause large-scale blackouts. Here, robustness measures how much performance is retained under extreme conditions, while resilience captures the grid's ability to re-stabilize and resume operation after recovery. Together, they describe a system's capacity to withstand and recover from perturbations.

**Robustness.** Robustness has been a central concept in control systems, which refers to the stability of the algorithm under uncertainties [48]. In MARL, the study of robustness relies on defining an uncertainty set $\mathcal{U}$ in the decision process. During the deployment of MARL policies, such uncertainty set $\mathcal{U}$ can be defined as a distribution over uncertainty realizations, where each $u \in \mathcal{U}$ represents a perturbation on observation [14], action [34, 40] or environments [38]. Let $\pi$ be a fixed policy, and $\rho_0$ the initial state distribution, we define the *robustness* of $\pi$ as:

$$J^{\text{robust}}(\pi) = \mathbb{E}_{u \sim \mathcal{U}} \left[ \mathbb{E}_{s_0 \sim \rho_0} \mathbb{E}_{\pi, u} \left[ \sum_{t=0}^\infty \gamma^t r_t \middle| s_0 \right] \right].$$

**Resilience.** When uncertainties cannot be handled by algorithms alone, robustness alone may become insufficient. In such cases, resilience—*i.e.*, *the ability of systems to recover from external shocks* [49]—becomes crucial. As illustrated in Fig. 2, resilience and robustness are complementary: while

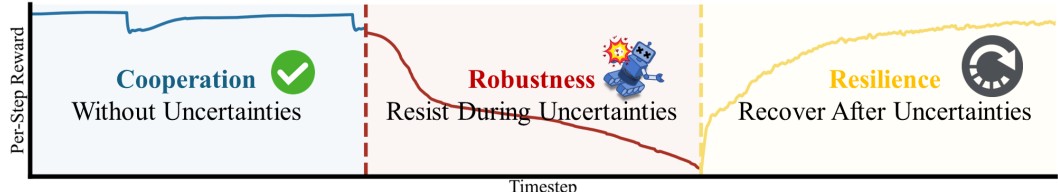

Figure 2: Relation between cooperation, robustness, and resilience under uncertainty. Cooperative MARL is trained without perturbations, but must be robust and resilient when they occur.

robustness allows a system to maintain functionality under small perturbations, resilience ensures recovery when perturbations exceed the limits of robustness [50]. This complementary relationship has been extensively explored across various fields, including control theory [21], ecology [22], economics [23], and complex networks [51].

Despite this, the MARL literature often conflates resilience with robustness, overlooking their distinct roles. For instance, [24] frame resilience as an inherent feature of robustness, while [25, 39] label their approaches as resilient MARL but ground their methodologies in robust RL. Similarly, [26] and [40] use the terms resilience and robustness interchangeably, failing to distinguish between the two. To explicitly define resilience as the capacity to recover from uncertainties, we propose a framework in which an episode begins at a perturbed state $s_u$ at timestep $t_u$, where the perturbation is induced by uncertainty $u \in \mathcal{U}$. From this perturbed state, the system evolves *without* additional uncertainties. Formally, let $s_u \sim \rho_u$ denote the initial state distribution, we define the resilience of $\pi$ as:

$$J^{\text{resilience}}(\pi) = \mathbb{E}_{u \sim \mathcal{U}} \left[ \mathbb{E}_{s_u \sim \rho_u} \mathbb{E}_\pi \left[ \sum_{t=t_u}^{\infty} \gamma^t r_t \,\middle|\, s_0 = s_u \right] \right].$$

We view resilience as a practically grounded form of generalization, emphasizing recovery and stability under unknown perturbations. In reinforcement learning, *generalization* measures a policy's ability to sustain performance across variations in initial states or environment dynamics, typically bounded or randomized during training [52]. In contrast, *resilience* concerns recovery after unexpected disturbances, characterized by a post-uncertainty state distribution $s^u$ that arises from real-world perturbations and often lies beyond standard generalization settings. Thus, resilience addresses a narrower yet practically crucial challenge for deployment and opens algorithmic directions toward policies that actively adapt or recover from disturbed states.

**Difference Between Robustness and Resilience.** We conduct two case studies to illustrate the distinction between robustness and resilience in multi-agent reinforcement learning. A system is *resilient but non-robust* if it recovers well from a variety of initial states, including those following adversarial perturbations, but fails under persistent uncertainties such as external forces or observation noise. In such cases, the policy may succeed at high-level task execution but lacks the fine-grained control needed for precise behavior, as small perturbations consistently mislead action selection. Conversely, a system is *robust but non-resilient* if it performs reliably under continuous uncertainty, yet fails to recover when initialized in perturbed or atypical states—often due to limited exposure to such states during training. In both cases, we observe evidence of *state aliasing*, where similar observation histories require subtly different responses, but the policy fails to distinguish them, resulting in degraded performance. Additional examples, results, visualizations, and analysis are provided in Appendix B.2.

## 4 Experiment Procedure

In this section, we describe the hyperparameters, types of uncertainties, environments, algorithms used for our evaluation. Our full experiment takes ∼230K GPU hours, measured in GTX 4090.

**Algorithms.** To align with the continuous control requirements of our real-world environments, we evaluate the impact of hyperparameters on three widely used MARL algorithms that support continuous control: MADDPG [1], MAPPO [3], and HAPPO [4], with HAPPO representing heterogeneous-agent-based methods. Our study focuses on commonly used algorithms to ensure broad applicability while prioritizing an in-depth analysis of hyperparameters. Our modular codebase enables easy integration of new algorithms via predefined interfaces, supporting future extensions.

**Hyperparameters.** For generality, we pick hyperparameters that are used by most methods. Specifically, we consider both general and algorithm-specific hyperparameters. General hyperparameters includes *network hidden size*, *discount factor*, *activation function*, *initialization method*, *neural network type*, *learning rate*, *critic learning rate*, *feature normalization*, *share parameters* and *early stop*. For MAD-DPG, algorithm-specific choices includes *N-step TD* and *exploration noise*. For MAPPO and HAPPO, algorithm-specific choices includes *entropy coefficient*, *Use GAE* and *Use PopArt*. The descriptions of each hyperparameters and the reasons for selecting them are deferred to Appendix A.1.1. The values of

| General hyperparameters | |
| --- | --- |
| Choices | Choice Range |
| Network Hidden Size | {64, **128**, 256} |
| Discount Factor ($\gamma$) | {0.9, 0.95, **0.99**} |
| Activation Function | {**ReLU**, Leaky_ReLU, SELU, Sigmoid, Tanh} |
| Initialization Method | {**Orthorgonal**, Xavier} |
| Neural Network Type | {**MLP**, RNN} |
| Learning Rate (LR) | {5e-5, **5e-4**, 5e-3} |
| Critic Learning Rate | {5e-5, **5e-4**, 5e-3} |
| Feature Normalization | {**True**, False} |
| Share Parameters | {True, **False**} |
| Early Stop | {True, **False**} |
| MADDPG Specific hyperparameters | |
| TD Steps (N Step) | {5, **20**, 25} |
| Exploratory noise | {0.001, 0.01, **0.1**, 0.5, 1} |
| MAPPO/HAPPO Specific hyperparameters | |
| Entropy Coefficient | {0.0001, 0.001, **0.01**, 0.1} |
| Use GAE | {**True**, False} |
| Use PopArt | {**True**, False} |

Table 1: General and algorithm-specific hyperparameters shared for all methods. Default choices are shown in bold font, which is shared by all algorithms and environments.

each hyperparameters are shown in Table. 1. We use bold font to denote the default choices, which are shared for all algorithms and environments. This leads to 15 different hyperparameters. In each time, we vary one different hyperparameters to test its effect, resulting in 34 models with different implementations. Other hyperparameters used in our experiments are listed in Appendix. A.1.2.

Table 2: Overview of evaluated environments.

| Environment | Task Type | Control Mode | Episode Len. | Engine/Source | Data Source | Challenge |
| --- | --- | --- | --- | --- | --- | --- |
| DexHand | Multi-robot Manipulation | Continuous | ∼80 | Isaac Gym | Real-world Robots | Precise Control |
| Quads | Multi-robot Navigation | Continuous | ∼1600 | OpenAI Gym | Real-world Robots | Long-range Task Assignment |
| Traffic | Network Control | Discrete | ∼1000 | SUMO | Real-world Traffic | Long-range Control |
| Voltage | Network Control | Continuous | ∼200 | IEEE Standard | Real-world Power Grid | Complex and Noisy Dynamics |

**Real-World Environments.** Unlike conventional benchmarks that consider solely simulation environments, our study incorporates four real-world environments encompassing diverse task types, control modes, episode lengths, simulation engines, data sources, and control challenges, as summarized in Table. 2. The first two environments, *Dexterous Hand Manipulation (Dexhand)* [53] and *Quadrotor Swarm Control (Quad)* [54] are grounded in real-world applications, allowing policies learned in simulation to be directly transferred to physical robots with the same dynamic. We use these environments for tasks that requires robustness in precise control. We select six tasks in each environment. The remaining two environments, *Intelligent Traffic Control (Traffic)* [55] and *Active Voltage Control (Voltage)* [56] are constructed from real-world data, ensuring high-fidelity replication of real-world dynamics. We use these environments for tasks that requires robustness in adaptive control when cooperators are unavailable. We select all three tasks with homogeneous observation and action space for *Traffic* and all three tasks for *Voltage*, resulting in 18 tasks in total. See detailed descriptions of environments and tasks in Appendix. A.2.

**Types of Uncertainties.** We evaluate the robustness and resilience of MARL algorithms under 13 distinct uncertainties spanning observation, action, and environment. For **observation uncertainties (obs)**, we consider three types: *Gaussian noise*, *greedy worst-case attacks* using white-box, gradient-based methods to generate perturbations [30], and *learned optimal attacks* via MARL [14, 15]. These uncertainties are applied either to all agents with a small perturbation budget ($\epsilon = 0.1$) or to a single agent with a large budget ($\epsilon = 0.2$), resulting in six scenarios. For **action uncertainties (act)**, we assess three types: *random policies*, *greedy worst-case policies* [34], and *learned optimal policies* [16]. Action perturbations are modeled as $\epsilon\hat{\pi} + (1 - \epsilon)\pi$, where $\hat{\pi}$ represents the perturbed policy. Similar to observation uncertainties, these are applied across all agents ($\epsilon = 0.1$) or to a single agent ($\epsilon = 0.2$), yielding six scenarios. For **environmental uncertainties (env)**, following [57, 58, 59, 60], we sample 50 rollouts uniformly from uncertainty sets over environmental hyperparameters (*e.g.*, mass, velocity) and report the worst-case result. In total, our study includes 6 observations, 6 actions, and 1 environmental uncertainty, totaling 13 distinct scenarios. We assess both robustness and resilience under these uncertainties, resulting in 27 evaluation settings: 1 cooperative baseline, 13 robustness evaluations, and 13 resilience evaluations. Further details are provided in Appendix. A.3.

**Experiment Procedures.** In each experiment, we first train all algorithms and tasks using default hyperparameters. Then, for each hyperparameter in Table. 1, we vary one setting at a time to create a

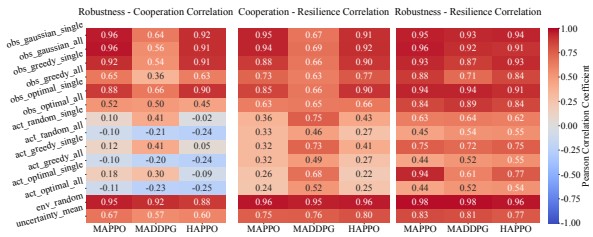
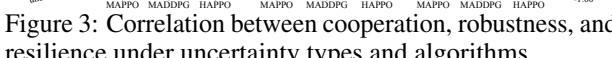

Figure 3: Correlation between cooperation, robustness, and resilience under uncertainty types and algorithms.

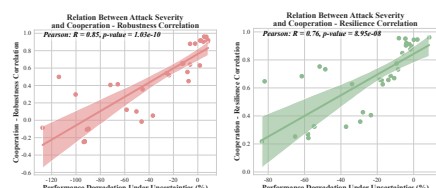

Figure 4: The extent cooperation is correlated to robustness and resilience linearly depends on the severity of the attack.

set of cooperative models. To evaluate robustness, we fix all cooperative models and measure their performance under uncertainties. Finally, we evaluate resilience by initializing a new episode from the perturbed state reached by cooperative models after encountering uncertainties. We then assess their performance from this state, measuring their ability to recover from uncertainties. We train all cooperative models using 5 random seeds, and report the robustness and resilience results on the same five seeds as the cooperative model. In total, our study includes 5 (random seeds) × 27 (uncertainty settings) × 18 (tasks) × 34 (hyperparameters) = 82620 experiments.

**Architecture.** For practitioners to evaluate the robustness and resilience in their own settings, our codebase allows users to easily integrate custom algorithms and environments by adhering to a well-defined interface. This flexible architecture ensures that users can seamlessly adapt our framework to their specific requirements. We defer these details to Appendix. A.4.

## 5 Experiment Results and Takeaways

In this section, we thoroughly analyze the data gained in our experiments and summarize our key findings as takeaways for MARL practitioners and researchers. A concise summarization of our finding is shown in Table. 3. To account for variations in reward scaling across tasks, we use normalizations for different analysis, with details in Appendix. B.1. We report statistically significant results to mitigate the randomness introduced by diverse test conditions, while noting that these findings reflect general trends and may not hold in every individual case.

Table 3: Recommended practices for handling different scenarios in MARL.

| Scenario | Suggestion | Reference |
|---|---|---|
| Small uncertainties | Optimize cooperation | Section.5.1 |
| Large uncertainties | Requires dedicated robustness/resilience strategies | Section.5.1 |
| Obs. uncertainties | MAPPO/HAPPO best | Section.5.1 |
| Action uncertainties | MADDPG best | Section.5.1 |
| Modalities | Evaluate obs/env/action separately | Section.5.2 |
| Agent scopes | Test both individual/global perturbations | Section.5.2 |
| Early stop | On | Section.5.3 |
| Critic LR | > Actor LR | Section.5.3 |
| Activation | Leaky ReLU | Section.5.3 |
| GAE | Off | Section.5.3 |
| PopArt | Off | Section.5.3 |
| Param sharing | On (homogeneous), Off (heterogeneous) | Section.5.3 |
| Exploration | High (MAPPO/HAPPO), Moderate (MADDPG) | Section.5.3 |

### 5.1 Are Cooperation, Robustness, and Resilience Correlated?

Our first evaluation investigates the relationship between cooperation, robustness, and resilience. We compute the Pearson correlation among these metrics to determine whether maximizing cooperation naturally leads to higher robustness and resilience. The results are shown in Fig. 3.

**1. The correlation of cooperation with robustness, and resilience depends on the severity of the attack.** At the first glance, robustness and resilience seems highly correlated with cooperation (Fig. 3), particularly under observation and environmental uncertainties. However, further analysis reveals that this correlation is strongly influenced by the severity of the uncertainty. Specifically, Fig. 4 illustrates the relationship between correlation strength and the percentage of performance degradation under varying uncertainties. The results show a clear linear trend: as the severity of the attack increases, the Pearson correlation between cooperation and robustness ($r = 0.85, p < .001$) and between cooperation and resilience ($r = 0.76, p < .001$) weakens significantly. This suggests that mild uncertainties in MARL can be countered by simply maximizing cooperative performance, but their impact becomes critical when perturbation is severe. Similarly, while robustness and

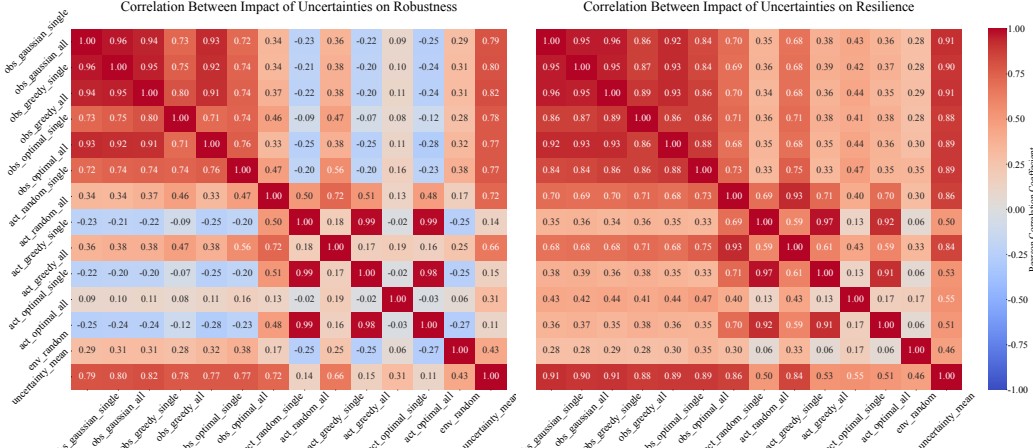

Figure 5: The self-correlation between 13 types of uncertainties in terms of robustness and resilience. Correlations in uncertainties differ significantly across modalities (observations, actions, environments) and scope of agents (applied to individual or all agents).

resilience seem correlated, such correlation is small under large uncertainties, indicating that these two properties behave independently and should not be treated interchangeably.

**2.  Linearity across uncertainty types, agent scopes, and attack strategies.** A closer look reveals strong correlations under random, greedy, and optimal attacks, as well as under observation uncertainty and perturbations affecting either single or all agents ($r \geq 0.6$, $p < .05$). In the case of environment uncertainty, correlations remain strong ($r \geq 0.85$) but lack statistical significance. Under action uncertainty, MADDPG exhibits greater robustness than MAPPO and HAPPO, resulting in weak overall correlations. When MADDPG is excluded, MAPPO and HAPPO show strong linearity with robustness under action uncertainty ($r \geq 0.69$, $p < .05$). See full results in Appendix. B.3.

**3. Algorithms exhibit similar overall robustness and resilience, but differ in sensitivity to uncertainty types.** While no significant differences are observed in overall robustness and resilience across algorithms, we find that MADDPG is more resilient to action uncertainties, whereas MAPPO and HAPPO perform better under observation uncertainties. This discrepancy stems from their training methods: MAPPO and HAPPO leverage a centralized critic, which improves sample efficiency and robustness against noisy observations. While these on-policy methods experience high randomness for max-entropy learning at the beginning of training, the policy becomes more deterministic as policy converge, making it still non-robust against action noise. As for MADDPG, the exploration noise is maintained throughout training. As training progresses, MADDPG continues to inject noise in the action space, which contributes to its greater robustness to action perturbations, especially in late-stage policies that retain some level of stochastic behavior.

**Takeaway.** Cooperation improves robustness and resilience under mild uncertainty, but this correlation weakens as attack severity increases. The phenomenon holds across most uncertainty types, agent scopes, and attack strategies. MADDPG is preferable for action uncertainties, while MAPPO and HAPPO are better suited for observation uncertainties.

## 5.2  Uncertainty Diversity Matters

We next examine the relationships among 13 types of uncertainties to determine whether robustness against a single uncertainty type generalize to all other types. To achieve this, we compute the Pearson correlation between all pairs of uncertainties, conditioned on robustness and resilience. The key findings are summarized below:

**1. Observation, environment, and action uncertainties are uncorrelated.** As shown in Fig. 5, compared to the 6 subtypes of uncertainties within observation and action categories, the three modalities of uncertainties—observation, environment, and action—exhibit significantly lower correlations. A one-way ANOVA confirms this effect is statistically significant for robustness($F(2, 153 = 9.53, p < .001$) and resilience ($F(2, 153) = 14.51, p < .001$). This indicates that robustness against one modality does not imply robustness against other modalities. Such phenomenon is observed in

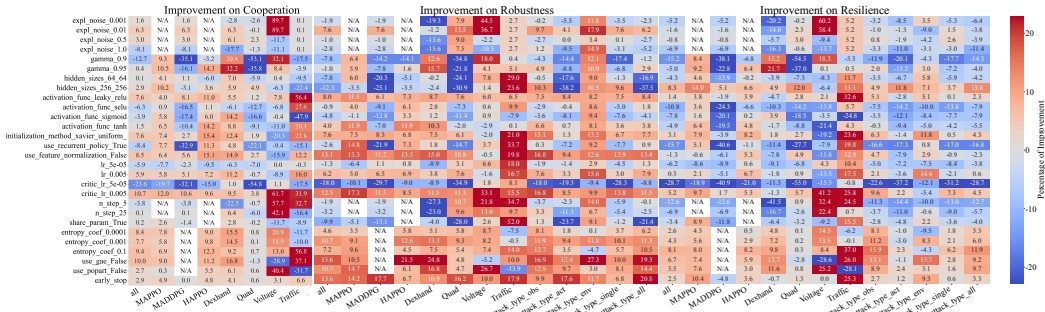

Figure 6: Percentage change in cooperation, robustness, and resilience caused by varying hyperparameters. A 5% winsorization is applied to limit extreme value within two standard deviations.

EIR-MAPPO [40], where policies trained to be robust against action uncertainties remain vulnerable to observation uncertainties.

**2. Uncertainties applied to individual agents and all agents are uncorrelated.** Similarly, we find that the correlations between uncertainties applied to individual agents and those applied to all agents are significantly different. A one-way ANOVA confirms this effect for robustness ($F(1, 142) = 4.36, p < .05$) and resilience ($F(1, 142) = 7.00, p < .05$). This explains the issue encountered in M3DDPG [34], where agents trained to be robust against small uncertainties applied uniformly across all agents fail when exposed to large uncertainties applied to individual agents.

**Takeaway.** Robustness and resilience in MARL can not generalize across uncertainty modalities or agent scopes. Trustworthy MARL systems must therefore evaluate against diverse types of uncertainty, and account for both individual and group-level perturbations.

### 5.3 What hyperparameters are effective?

In this section, we examine the impact of hyperparameters on cooperation, robustness, and resilience. This is motivated by our preliminary study, where two-way ANOVA shows hyperparameters have a stronger effect than MARL algorithms in 9 of 18 tasks ($p < 0.001$), highlighting the need to quantify their influence on trustworthy MARL. Due to space constraints, detailed results are presented in Appendix. B.4. To identify broadly effective hyperparameters, we assign equal weight to each task and apply 5% winsorization to limit extreme values within two standard deviations. Figure. 6 reports the percentage change in cooperation, robustness, and resilience due to change of hyperparameters. The results are summarized as follows.

**1. Early stopping is generally effective.** During MARL training, after the convergence of cooperative performance, robustness and resilience may continue to evolve. As shown in Fig. 7, we implement an early stopping criterion that saves models based on their combined performance in cooperation, robustness, and resilience. This ensures the saved model performs no worse than the model at the final training round. In contrast, existing codebases [42, 3] save model in the final timestep, which is significantly worse than our early stopping criterion ($t(161) = 6.16, p < .001$, paired t-test).

**2. Use a higher learning rate (LR) for the critic.** Using a learning rate for the critic higher than the actor often improves cooperation, robustness, and resilience simultaneously ($t(161) = 10.02, p < .001$, paired t-test). This aligns with theoretical studies on the convergence of actor-critic algorithms using two-timescale stochastic optimization [61, 62, 63]. Stable convergence is achieved where a faster LR for the critic ensures it is essentially equilibrated, while the actor's policy updates at a slower rate and is quasi-static. Despite this well-established theoretical insight, most MARL codebases set equal LRs for the actor and the critic [42, 3].

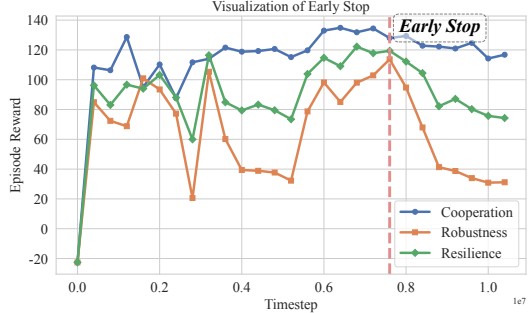

Figure 7: The effect of early stop. While cooperation performance converge, robustness and resilience continue to evolve. We use early stop to select the best model.

**3. Use Leaky ReLU for activation.** Leaky ReLU consistently outperforms ReLU in cooperation, robustness, and resilience ($t(161) = 6.31, p < .001$, paired t-test). For cooperation, Leaky ReLU avoids the dead neuron problem inherent in ReLU, enabling better representation learning. For robustness and resilience, ReLU's dead neurons can become unexpectedly activated under distribution shifts caused by uncertainties, introducing errors into the network. In contrast, Leaky ReLU maintains stable gradients, preventing such issues.

**4. GAE and PopArt do not improve performance.** Despite being standard in modern RL and MARL, GAE ($t(107) = 7.44, p < .001$) and PopArt ($t(107) = 4.84, p < .001$) reduce cooperation, robustness, and resilience in our benchmark (paired t-test). To verify this counterintuitive finding, we conduct additional experiments on several widely used MARL benchmarks (Appendix B.5), confirming that while effective in Multi-Agent MuJoCo, these methods have limited impact in MPE and SMAC. We hypothesize that GAE performs well in MuJoCo due to its dense and stable reward structure, whereas in real-world environments, sparse and highly variable rewards amplify bootstrapping errors for GAE. See Appendix. B.6 for an example. As for PopArt, its normalization benefits may be unnecessary when reward scales remain stable, leading to inconsistent effectiveness across different tasks.

**5. Parameter sharing does not improve overall performance.** Although parameter sharing is highlighted by MAPPO as a key technique for its success [3], our findings show that it often reduces cooperation, robustness, and resilience ($t(161) = 7.01, p < .001$, paired t-test). Further evaluation on MPE, SMAC and Multi-Agent Mujoco reveals that improvements from parameter sharing are limited to SMAC tasks (see Appendix B.5 for details). While parameter sharing improves sample efficiency, it hinders the learning of agent-specific features, particularly in environments like Voltage and Traffic, where heterogeneous nodes require distinct strategies. Additionally, parameter sharing assumes that all agents follow the same policy, making the shared policy more vulnerable to attacks when other agents are impacted by uncertainties. Our claim can be reinforced by HATRPO [4], where it construct a negative example showing parameter sharing brings exponentially worse performance.

**6. Exploration benefits MAPPO and HAPPO.** Increasing exploration improves the cooperation, robustness and resilience of PPO-based methods ($t(107) = 6.64, p < .001$, paired t-test). The entropy bonus in these methods naturally arises as a consequence of the optimal solution in maximum entropy RL [64], and provably enhances robustness against environmental uncertainties [65]. Greater exploration also increases state space coverage, which enhances resilience. Our results confirm these benefits in MARL. For MADDPG, however, it does not enjoy such theoretical advantages and the impact of increased exploration is task-dependent. These differences reflect distinct exploration schemes. In MAPPO/HAPPO, entropy regularization acts as a soft constraint, which encourage exploration at the beginning, while the policy can still converge to a relatively deterministic policy to reach optimal cooperation. In contrast, MADDPG relies on additive action noise that typically persists throughout training. With large exploratory noise, a well-trained policy can be perturbed to low-value or unstable regions, degrading policy quality.

**7. Discussions on task-specific choices.** We do not find universal preference for learning rates, hidden state sizes, discount factors, use of RNN, or network initialization methods. However, these choices may significantly impact performance and should be tuned based on the task applied.

**Takeaways.** We recommend following our guidance when applying MARL to new environments with uncertainties. Notably, several findings challenge established assumptions in RL/MARL literature, including the use of parameter sharing, GAE, and PopArt. These discrepancies likely stem from the narrow task and environment focus of previous studies, highlighting the task-dependent nature of MARL performance.

## 5.4 Improving Robustness and Resilience

In this section, we demonstrate that robustness and resilience can be significantly improved through hyperparameters alone. For each task, we identify the best set of hyperparameters that maximize the combined performance of cooperation, robustness, and resilience. Again, we apply a 5% winsorization to suppress outliers. As shown in Fig. 8, optimizing hyperparameters during training leads to substantial performance improvements, with increases of 52.60% in cooperation, 34.78% in robustness, and 60.34% in resilience.

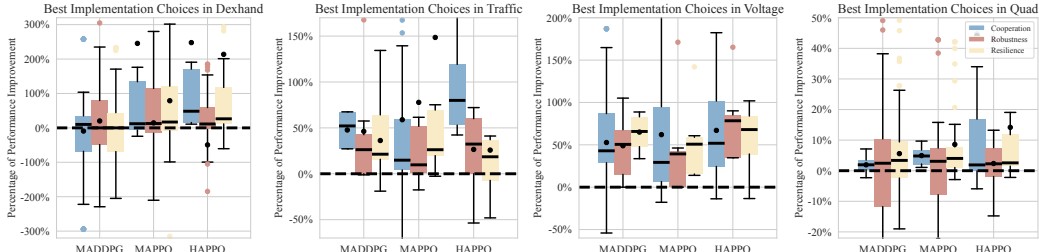

Figure 8: Percentage improvement for each environments using the best tricks for each task. Cooperation, robustness and resilience can be obtained altogether by simply tuning hyperparameters.

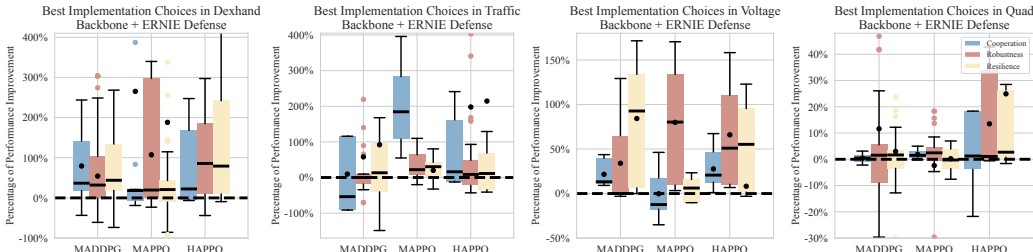

Figure 9: Generalization of the effect of hyperparameter tuning. We report percentage improvement on ERNIE defense with diverse backbones and environments.

To show the generality of hyperparameter tuning, we train ERNIE [66], a general-purpose robust MARL method on all backbones, using default hyperparameters and our selected best hyperpararmeters on non-robust backbones above. As shown in Fig. 9, the improvement of hyperparameter generalize to robust MARL methods under various backbones and environments, with an average improvement of 89.43% on cooperation, 65.83% on robustness, 82.96% on resilience.

Post-hoc analysis of the interactions among hyperparameters was conducted using ordinary least squares regression, a statistical method that quantifies the significance of each factor's contribution. Results show it is beneficial to use parameter sharing for homogeneous agents ($p < .05$), smaller discount factors for short episodes ($p < .001$), higher critic LR in policy gradient methods ($p < .05$) and always use early stop ($p < .001$). See full results in Appendix. B.7.

## 6  Conclusions

In this paper, we conducted a large-scale empirical study on the robustness and resilience of MARL, yielding several meaningful insights: (1) Optimizing for cooperation improves robustness and resilience under mild uncertainty, but this relationship weakens as perturbations intensify, with performance differing across algorithms and uncertainty types. (2) Robustness and resilience do not transfer across uncertainty modalities or agent scopes; a policy robust to action noise affecting all agents may still fail under observation noise targeting a single agent. (3) Hyperparameter tuning plays a central role in trustworthy MARL. Surprisingly, widely used practices like parameter sharing, GAE, and PopArt can degrade robustness, while strategies such as early stopping, higher critic learning rates, and Leaky ReLU offer consistent improvements. By optimizing hyperparameters only, we observe substantial improvement in cooperation, robustness and resilience across all MARL backbones, with the phenomenon also generalizing to robust MARL methods across these backbones. A limitation of our paper is the focus of MARL algorithms based on policy gradient, since many environments requires continuous control. This can be mitigated by integrating new algorithms into our codebase, which supports custom environments and algorithm integration.

## Acknowledgements

This work was supported by National Natural Science Foundation of China (62476018).

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

# Appendix for "Empirical Study on Robustness and Resilience in Cooperative Multi-Agent Reinforcement Learning"

## A Appendix for Experiment details

### A.1 Additional Details on hyperparameters

In this section, we first discuss the key hyperparameters that significantly affect the performance of the MARL algorithms evaluated in our benchmark. Next, we provide an overview of all hyperparameter settings used across the different algorithms and highlight configurations specific to each algorithm.

#### A.1.1 Introduction on hyperparameters

The following subsections provide a brief description of the critical hyperparameters and settings that impact the implementation of the MARL algorithms. As no universal set of hyperparameters exists for multi-agent reinforcement learning (MARL) algorithms, our selections are informed by established practices in reinforcement learning, with particular reference to the codebases of MAPPO [3] and HAPPO [67]. These codebases provide carefully designed hyperparameter configurations that perform well across environments such as MPE, SMAC, and MAMujoco, though the optimal settings vary by task. Building on this, we conducted a pilot study to identify a subset of hyperparameters that consistently work well across most of our real-world tasks, while adopting alternative configurations for the remaining parameters as needed.

1. **Network Hidden Sizes:** This refers to the size of the actor networks and critic networks used in the MARL algorithms, which determines the number of neural units per layer. The choices are {64, 128, 256}, with the default value being 128.

2. **Discount Factor:** The standard parameter in RL, which controls the importance of future rewards during training, with a choice set of {0.9, 0.95, 0.99}. The default value is 0.99.

3. **Activation Function:** Various activation functions in neural networks, including ReLU, Leaky ReLU, SELU, Sigmoid, and Tanh. The default function is ReLU.

4. **Initialization Method:** The initialization of network weights is conducted using either Orthogonal or Xavier initialization. Orthogonal initialization is the default choice.

5. **Neural Network Type:** This parameter denotes the network architecture of actor and critic in MARL algorithms, with options including MLP (Multi-Layer Perceptron) and RNN (Recurrent Neural Network). The default architecture is MLP.

6. **Learning Rate:** The learning rate for training the actor network is selected from the set {5e-5, 5e-4, 5e-3}, with a default value of 5e-4.

7. **Critic Learning Rate:** The critic network learning rate is chosen from the same set as the actor network, {5e-5, 5e-4, 5e-3}, with 5e-4 as the default value.

8. **Feature Normalization:** This applies Layer Normalization to standardize the features of each sample, ensuring they have a mean of 0 and a variance of 1. This helps stabilize training and improve convergence. Feature normalization is enabled by default.

9. **Share Parameters:** This parameter specifies whether multiple agents share the same neural network parameters for their policy functions. The default setting is `False`. Note that HAPPO inherently assumes heterogeneous agents, and do not support parameter sharing.

10. **TD Steps (MADDPG):** TD steps determine the number of steps used in Temporal Difference (TD) learning for updating the value estimates. The choices available are {5, 20, 25}, with a default value of 20.

11. **Exploratory Noise (MADDPG):** This parameter controls the scale of noise added to the actions generated by the actor network, in order to encourage exploration during training. The options available are {0.001, 0.01, 0.1, 0.5, 1}, with a default value of 0.1.

12. **Entropy Coefficient (MAPPO/HAPPO):** Controls the entropy regularization strength in the MAPPO/HAPPO objective, which affects exploration. The selection set is {0.0001, 0.001, 0.01, 0.1}, with a default value of 0.01.

13. **Use GAE (MAPPO/HAPPO):** Whether Generalized Advantage Estimation (GAE) is used for estimating the advantage of a state-action pair. The default is `True`.

14. **Use PopArt (MAPPO/HAPPO):** PopArt is a normalization technique used to rescale value targets dynamically to improve training stability. The default choice is `True`.

15. **Early Stop:** We use early stop as a model selection criterion. At each evaluation period, we evaluate the cooperation, robustness and resilience of algorithms, and save the model with maximum cooperation + robustness + resilience. Baseline model save the model at their last round of evaluation. The default choice is `False`.

### A.1.2 Other Default Hyperparameters

The following table summarizes the default hyperparameters used across the various MARL algorithms. We first describe the general hyperparameters that are common across all algorithms, followed by algorithm-specific parameters.

Table 4: Hyperparameters for all MARL algorithms.

| Number | Hyperparameter | MAPPO | HAPPO | MADDPG |
|---|---|---|---|---|
| **General Hyperparameters** | | | | |
| 1 | Neural network type | MLP | MLP | MLP |
| 2 | Network hidden sizes | [128, 128] | [128, 128] | [128, 128] |
| 3 | Activation function | ReLU | ReLU | ReLU |
| 4 | Discount factor ($\gamma$) | 0.99 | 0.99 | 0.99 |
| 5 | Actor learning rate | 0.0005 | 0.0005 | 0.0005 |
| 6 | Critic learning rate | 0.0005 | 0.0005 | 0.0005 |
| 7 | Feature normalization | True | True | True |
| 8 | Share parameters | False | False | False |
| 9 | Weight initialization method | Orthogonal | Orthogonal | Orthogonal |
| 10 | Recurrent layers (if use RNN) | 1 | 1 | 1 |
| 11 | Use max gradient norm | True | True | True |
| 12 | Max gradient norm | 10.0 | 10.0 | 10.0 |
| 13 | Linear learning rate decay | False | False | False |
| **MAPPO and HAPPO Shared Hyperparameters** | | | | |
| 14 | Entropy coefficient | 0.01 | 0.01 | N/A |
| 15 | Use PopArt | True | True | N/A |
| 16 | Use GAE | True | True | N/A |
| 17 | GAE parameter ($\lambda$) | 0.95 | 0.95 | N/A |
| 18 | PPO epochs | 5 | 5 | N/A |
| 19 | Critic epochs | 5 | 5 | N/A |
| 20 | Clip parameter | 0.05 | 0.1 | N/A |
| 21 | Use clipped value loss | True | True | N/A |
| 22 | Value loss coefficient | 1.0 | 1.0 | N/A |
| 23 | Action aggregation | Product of probabilities | Product of probabilities | N/A |
| 24 | Use huber loss | True | True | N/A |
| 25 | Huber delta | 10.0 | 10.0 | N/A |
| **Algorithm-Specific Hyperparameters** | | | | |
| 26 | Exploration noise | N/A | N/A | 0.1 |
| 27 | TD steps | N/A | N/A | 20 |
| 28 | Buffer size | N/A | N/A | 5000 |
| 29 | Batch size | N/A | N/A | 1000 |
| 30 | Polyak averaging factor | N/A | N/A | 0.005 |

Table 4: Hyperparameters for all MARL algorithms.

| Number | Hyperparameter | MAPPO | HAPPO | MADDPG |
|--------|----------------|-------|-------|--------|
| 31 | Warmup steps | N/A | N/A | 50000 |
| 32 | Update per training step | N/A | N/A | 1 |
| 33 | Final activation function | N/A | N/A | Tanh |

## A.2 Additional Details on Environments

Our benchmark study is grounded on four real-world environments, as illustrated in Fig. 10. The first two, *Dexterous Hand Manipulation (Dexhand)* [53] and *Quadrotor Swarm Control (Quad)* [54], are rooted in real-world robotics, enabling seamless policy transfer from simulation to physical systems. In dexterous hand control and quadrotor swarm control, the MARL algorithm needs to control the low-level actions made by robot, where a small variation might leads to large change in robot actions, requiring robust control in precise manipulations. Each environment comprises six tasks. The other two environments, *Intelligent Traffic Control (Traffic)* [55] and *Active Voltage Control (Voltage)* [56], are based on real-world data, are derived from real-world data, providing high-fidelity simulations of complex systems. In intelligent traffic control and active voltage control, each agents in MARL algorithm are highly responsible for the performance of the overall system, requiring adaptive control with uncertainties. These environments feature four tasks and three tasks, respectively, resulting in a total of 19 tasks across all environments (6 for Dexhand + 6 for Quad + 3 for Traffic + 3 for Voltage = 18 tasks). The following subsections offer a more detailed description of the environments and tasks included in our benchmark.

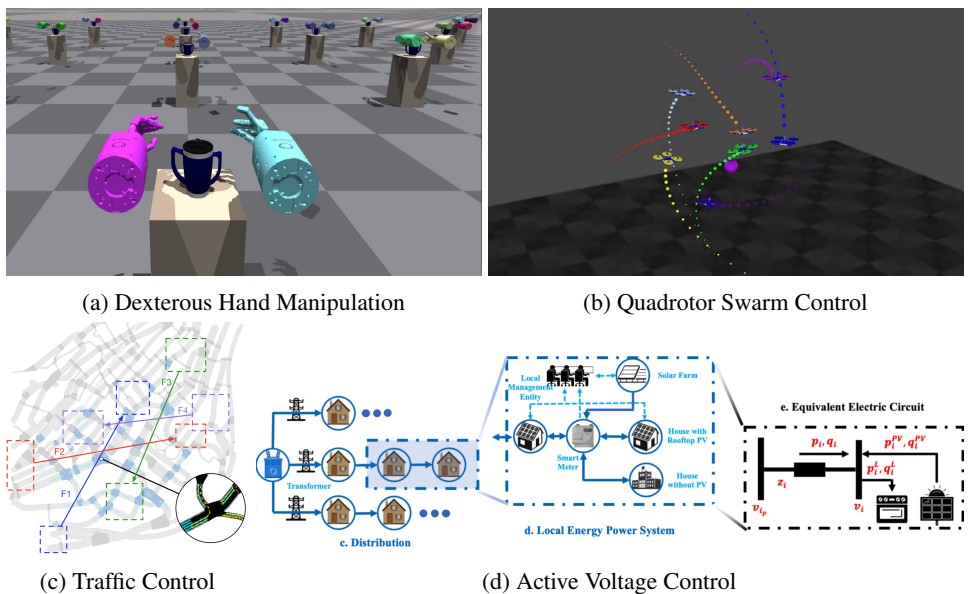

(a) Dexterous Hand Manipulation  (b) Quadrotor Swarm Control

(c) Traffic Control  (d) Active Voltage Control

Figure 10: Illustration of four real-world environments in the benchmark.

### A.2.1 Dexterous Hand Manipulation

The Bimanual Dexterous Hand Manipulation environment (DexHand) is a comprehensive simulation platform designed to mimic human dexterity through reinforcement learning. Using two anatomically realistic Shadow Hands, each with 24 degrees of freedom (DoF), it tackles tasks ranging from basic object manipulation to advanced bimanual coordination like stacking, catching, and reorientation. These tasks are designed to match different levels of human motor skills according to cognitive science literature, and demand precise control, adaptability, and synchronized multi-agent cooperation. Powered by the Isaac Gym simulator, Bi-DexHands combines physical realism with challenging, multi-agent tasks, presenting a high level of difficulty for learning and mastering these tasks.

In this work, six typical tasks are considered for the deployment of all the MARL algorithms and attacks in our benchmark. These tasks include: *PushBlock*, *SwingCup*, *Re-Orientation*, *BlockStack*, *HandCatchUnderarm* and *HandCatchOver2Underarm*. Below, we provide a brief description of these 6 tasks. For more details, we direct readers to the official project page of Bi-DexHands `https://github.com/PKU-MARL/DexterousHands`.

- **PushBlock**: This task requires both hands to touch the block and push it forward.

- **SwingCup**: This task requires two hands to hold the cup handle and rotate it 90 degrees.

- **Re-Orientation**: This task involves two hands and two objects. Each hand holds an object and we need to reorient the object to the target orientation.

- **BlockStack**: This task involves dual hands and two blocks, and we need to stack the block as a tower.

- **HandCatchUnderarm**: In this problem, two shadow hands with palms facing upwards are controlled to pass an object from one palm to the other. What makes it more difficult is that the hands' translation and rotation degrees of freedom are not frozen but are added into the action space.

- **HandCatchOver2Underarm**: In this task, the object needs to be thrown from the vertical hand to the palm-up hand.

### A.2.2 Quadrotor Swarm Control

The Quadrotor Swarm Control (Quad) environment is designed to train RL policies for controlling quadrotor swarms. In this environment, each quadrotor is controlled by an individual RL agent, enabling the development of sophisticated swarm behaviors. These include performing tight maneuvers in formation, avoiding collisions, adapting to dynamic obstacles, and collaborating in pursuit-evasion tasks. The environment simulates realistic quadrotor flight physics, accurately capturing the complex dynamics of aerial movement and multi-agent interactions. This ensures that the policies learned within the simulator can be generalized to real-world systems.

The environment includes several key scenarios to assess the quadrotors' abilities in different contexts. These tasks focus on the control, coordination, and safety of the swarm under various challenging conditions. Our benchmark experiments consist of six tasks: *static_diff_goal*, *static_same_goal*, *o_static_same_goal*, *o_random*, *o_swap_goals*, and *swarm_vs_swarm*, where *o_* denotes environments with a moving obstacle. A more detailed description of these six tasks is provided below, and additional information can be found on the official project page at `https://sites.google.com/view/swarm-rl`.

- **static_diff_goal**: In this task, quadrotors are trained to fly in close formation while maintaining a cohesive structure and avoiding collisions. The target formation, which remains fixed throughout the episode, can take various geometric shapes (e.g., 2D grid, circle, cylinder, and cube). The separation $r$ between goals within the formation is randomly chosen.

- **static_same_goal**: This is a special case of the *static_diff_goal* scenario, where the separation between goals is zero ($r = 0$). In this case, the goal locations for all quadrotors coincide, creating a dense formation with a high probability of collisions. This task requires enhanced coordination and decentralized control to ensure safe flight compared to the *static_diff_goal* task.

- **o_static_same_goal**: This task is similar to the *static_same_goal* task but takes place in an environment with dense cylindrical obstacles. The quadrotors must maintain their formation while avoiding collisions with the obstacles.

- **o_random**: In this task, each quadrotor's position in the formation is randomly sampled within the environment, which contains dense cylindrical obstacles. Every quadrotor needs to reach its own target position as fast as possible.

- **o_swap_goals**: This is a kind of dynamic formation control task. Given a predefined formation, the target positions of the quadrotors are randomly swapped multiple times during the episode. Additionally, the quadrotors must avoid collision with the dense obstacles in the environment.

- **swarm_vs_swarm**: In this task, the swarm is split into two groups, then the target formations of quadrotors in these two groups are swapped several times per episode, which requires two teams of quadrotors to fly through each other while avoid head-on collisions at high speed.

### A.2.3 Intelligent Traffic Control

The Intelligent Traffic Control (Traffic) environment focuses on training MARL agents to manage networked traffic systems. It supports various real-world traffic tasks, including Adaptive Traffic Signal Control (ATSC) and Cooperative Adaptive Cruise Control (CACC). In this environment, each agent represents either a traffic signal or a vehicle. Agents adjust their policies based on local observations and messages from neighboring agents, aiming to optimize overall traffic flow and safety. For ATSC tasks, agents control traffic signals at intersections, dynamically adjusting green/red light timings in response to local traffic conditions (e.g., vehicle density, wait times) and messages from adjacent intersections. In CACC tasks, agents simulate cooperative vehicle control, adjusting their speeds to either follow or close the gap with a leading vehicle, ensuring smooth, coordinated traffic flow.

This environment is capable of simulating both synthetic and real-world traffic networks, offering challenging scenarios for traffic signal and cooperative vehicle control tasks. Its versatility makes it an ideal platform for capturing the complexities of real-world traffic systems, such as partial observability, non-stationary dynamics, and decentralized control. In our benchmark, we evaluate performance across four tasks within this environment: *ATSC Grid*, *ATSC Monaco*, *CACC Catch-up*, and *CACC Slow-down*. A detailed description of these tasks can be found in the original paper [55] and on the official project page `https://github.com/cts198859/deeprl_network`.

- **ATSC Grid**: This task simulates traffic signal control within a synthetic 5×5 grid-based traffic network. Each intersection in the grid is controlled by an agent, which adjusts its signal timings based on local traffic conditions (e.g., vehicle density, wait times) and messages from neighboring agents. The task aims to optimize traffic flow and reduce congestion in this synthetic environment.

- **ATSC Monaco**: This task models adaptive traffic signal control in a real-world 28-intersection network, specifically based on the traffic system of Monaco city. Agents control traffic signals in this dynamic environment, addressing complex traffic patterns and interactions between intersections. The learned policies must effectively manage these complexities, focusing on reducing congestion, enhancing traffic throughput, and ensuring safe driving conditions. However, this task requires agents to have heterogeneous observation and action spaces. We thus omit this in our evaluation.

- **CACC Catch-up**: In this task, agents simulate CACC systems in a network of 8 vehicles. The goal is for each vehicle to follow and "catch up" with a leading vehicle by adjusting its speed based on the relative position and velocity of the vehicle ahead. This task emphasizes the importance of coordination between agents to maintain efficient vehicle spacing and flow.

- **CACC Slow-down**: This task simulates a CACC system in which 7 agents follow a leading vehicle and slow down appropriately. Agents adjust their speeds based on the behavior of the leading vehicle, ensuring safe distances while reducing speed when necessary. This scenario highlights the challenges of safely managing the traffic flow in situations where deceleration is required.

### A.2.4 Active Voltage Control

The Active Voltage Control (Voltage) environment provides a simulation platform for training RL policies to manage voltage in power distribution networks. It specifically targets the challenges posed by the integration of distributed energy resources, such as rooftop photovoltaics (PVs), into power grids, where excess power generation can cause voltage fluctuations. These fluctuations may exceed acceptable grid limits, as outlined by [68, 69], necessitating effective voltage regulation.

The primary objective of the Voltage environment is to mitigate these voltage fluctuations by controlling the reactive power generated by PV inverters. Each agent in the system corresponds to a PV inverter, which adjusts the reactive power to regulate the voltage at its respective bus. However, since the voltage at each bus is influenced by the power at all other buses, and not all buses are equipped

with PV inverters, agents must collaborate to ensure that the voltage across the entire network remains within a safe range. Given that each agent has limited visibility and can only observe the state of the local zone, the problem is naturally framed as a Decentralized Partially Observable Markov Decision Process (Dec-POMDP), requiring coordinated decision-making under uncertain conditions.

The Voltage environment includes three distinct scenarios based on real public data, each with different scales, ranging from small systems with 6 agents to larger systems with 38 agents. These scenarios vary in complexity, and our benchmark tests the robustness and scalability of multiple MARL algorithms across all three. The scenarios are referred to as the *33-Bus System*, *141-Bus System*, and *322-Bus System*. A brief overview of each follows, with further details regarding the configurations and complexities available in the original paper [56]:

- **33-Bus System**: This scenario models a small-scale distribution network with 6 PV agents, which cooperate to regulate the voltage across 32 loads distributed over 4 regions. The challenge is to maintain voltage within a safe range while minimizing power loss in a system with relatively fewer agents and simpler interactions.

- **141-Bus System**: This scenario involves a medium-scale distribution network with 22 PV agents that must coordinate to control voltage across 84 loads in 9 regions. The agents must manage more complex interactions and larger variations in power generation, requiring advanced coordination strategies.

- **322-Bus System**: This scenario simulates a large-scale distribution network with 38 agents controlling voltage across 337 loads in 22 regions. This complex environment emphasizes scalability and robustness, as agents must manage a larger network with greater uncertainty. Coordinating actions efficiently to maintain voltage stability is critical, despite the challenges posed by a large number of loads and agents, along with more dynamic power fluctuations. This scenario tests the performance and adaptability of MARL algorithms in large-scale control tasks.

## A.3 Additional Details on Type of uncertainties

In this section, we detail the type of observation, action and environment uncertainties considered in our paper.

### A.3.1 Observation uncertainty

Observation uncertainty arises when agents' sensing capabilities are flawed, inaccurate, or maliciously perturbed by attackers [29, 14, 15]. We consider the following types of uncertainties:

- **Gaussian noise.** Gaussian noise is ubiquitous in real-world systems, arising from sources such as thermal fluctuations, sensor inaccuracies, and environmental errors. While simple and relatively easy to mitigate, Gaussian noise serves as an essential baseline for our benchmark. Depending on the settings, we either apply small perturbations to all agents or larger perturbations to a single agent, denoted as *Gaussian-all* and *Gaussian-single*, respectively.

- **Greedy worst-case attacks.** Early research in robust RL and MARL introduced heuristic-based attacks to exploit vulnerabilities in agents' policies at individual timesteps. These methods specify a detrimental policy distribution and use gradient-based approaches, such as PGD [70], to generate observation perturbations that mislead agents. For instance, [29] proposed maximizing the KL divergence between the original policy and the perturbed policy at a given timestep. Other works explored alternative heuristics [71, 72, 73]. SA-MDP [30] formalized this problem as a state-adversarial MDP and proved the existence of a worst-case adversary. In our study, we adopt the Maximal Action Difference (MAD) attack from SA-MDP [30], which generalizes [29] by perturbing observations to maximize the KL divergence, $\max_{\hat{s}} D_{KL}(\pi(\cdot|s)||\pi(\cdot|\hat{s}))$, where $\hat{s}$ is the perturbed state. For MARL, as only observations are available, perturbations are applied to the observation space. Attacks targeting all agents are denoted as *Greedy-all*, while attacks targeting a single agent are denoted as *Greedy-single*.

- **Optimal learned attacks.** Moving beyond heuristic approaches, subsequent research developed adversarial policies using RL to optimize long-term attack efficacy. ATLA [74]

pioneered this area by learning a policy $\pi(\hat{s}|s)$ to generate perturbed states from the current state. PA-AD [75] improved ATLA by introducing a two-step process: first learning a target action for the perturbation to induce, and then applying gradient-based methods to craft state perturbations that guide the policy towards the target action. This approach is advantageous because actions often have lower dimensionality than states, simplifying the RL learning process. Similar to greedy attacks, we denote attacks against all agents as *Optimal-all* and those targeting a single agent as *Optimal-single*.

### A.3.2 Action uncertainty

Action uncertainties are a critical area of study in MARL because, in multi-agent systems, individual agents often take actions during deployment that deviate from optimal actions defined in simulation environments. These deviations can arise from various factors, such as external forces causing abrupt action changes, slight mismatches in robot weight compared to simulation models, or loss of control due to adversarial interference. We consider the following types of action uncertainties in MARL:

- **Random policies.** Agents may adopt random policies due to actuator errors, defaulting to random actions under limited computational resources, communication failures, or power fluctuations. To evaluate the robustness of MARL systems, we use random actions as a baseline. Depending on the setting, all agents may take random actions with a small probability, denoted as *Random-all*, or a single agent may take random actions with a large probability, denoted as *Random-single*.

- **Greedy worst-case policies.** Extending the concept of greedy worst-case perturbations in observations, we define greedy worst-case policies as those leading to the worst-case outcomes. Unlike observation perturbations, policy manipulations directly alter the action distribution, allowing for more targeted interventions. When the Q-function is available, we perturb policies to take actions that minimize the Q-value. If the Q-function is unavailable, policies are perturbed to take actions with the lowest probability. Depending on the setting, all agents may adopt greedy worst-case policies with a small probability, denoted as *Greedy-all*, or a single agent may do so with a large probability, denoted as *Greedy-single*.

- **Optimal learned policies.** Since policy perturbations naturally align with the RL/MARL framework, it is feasible to train a parameterized worst-case agent to generate perturbed policies. [32] formulated this as an action-robust MDP and used RL algorithms to learn worst-case perturbations. [16] studied this in two-agent zero-sum games, demonstrating how adversarial policies can easily exploit opponents. In MARL, [76, 40] learned worst-case policies to minimize cooperative rewards. Depending on the setting, all agents may take learned worst-case policies with a small probability, denoted as *Optimal-all*, or a single agent may do so with a probability of 1, denoted as *Optimal-single*.

### A.3.3 Environment uncertainty

Environment uncertainty has been a critical area of research since the inception of the robust RL framework [77, 78]. Given the inevitable discrepancies between simulation environments and real-world conditions, addressing uncertainties in environmental dynamics is a long-standing challenge in RL and MARL. In MARL, theoretical studies on this topic have been conducted by [38, 19], though these works lack empirical evaluation. To systematically evaluate the impact of environment uncertainties in MARL, we adopt approaches commonly used in RL [57, 58, 59, 60]. Specifically, we define uncertainty sets over key environment parameters such as velocity and mass to simulate potential environmental variations. The uncertainties we use are listed in Table. 5.

The following provides an explanation of the meanings of these uncertainty sets within their respective environments:

- **Quadrotor Swarm Control**:
    - **quads_collision_hitbox_radius**: This parameter defines the effective radius used for collision detection in quadrotor swarms, ranging from $[1.5, 2.5]$. When two quadrotors come within this distance, they are considered to have collided, prompting evasive maneuvers to ensure the swarm's safety.

Table 5: Uncertainty sets for environment uncertainties

| Environment | Uncertain Set | Range |
|---|---|---|
| Quadrotor Swarm Control | quads_collision_hitbox_radius | $[1.5, 2.5]$ |
| | quads_collision_falloff_radius | $[0.7, 1.3]$ |
| | quads_obst_density | $[-0.2, 0.2]$ |
| | quads_obst_size | $[0.7, 1.3]$ |
| | quads_collision_reward | $[-0.3, 0.3]$ |
| | quads_collision_smooth_max_penalty | $[7, 13]$ |
| | quads_episode_duration | $[10, 20]$ |
| | quads_obst_collision_reward | $[-0.3, 0.3]$ |
| | replay_buffer_sample_prob | $[0.5, 1]$ |
| Active Voltage Control | pv_scale | $[0.7, 1.3]$ |
| | demand_scale | $[0.7, 1.3]$ |
| | v_upper | $[0.705, 1.305]$ |
| | v_lower | $[0.65, 1.25]$ |
| | q_weight | $[0, 0.2]$ |
| | action_bias | $[-0.5, 0.5]$ |
| | action_scale | $[0.4, 1.2]$ |
| | demand_scale | $[0.5, 1.5]$ |
| Dexterous Hand Manipulation | startPositionNoise | $[-0.005, 0.005]$ |
| | transition_scale | $[0.2, 0.8]$ |
| | orientation_scale | $[-0.2, 0.4]$ |
| | rotRewardScale | $[0.7, 1.3]$ |
| | startRotationNoise | $[-0.5, 0.5]$ |
| | resetPositionNoise | $[-0.3, 0.3]$ |
| | resetRotationNoise | $[0.0, 0.01]$ |
| | resetDofPosRandomInterval | $[0, 0.4]$ |
| | resetDofVelRandomInterval | $[-0.2, 0.2]$ |
| Intelligent Traffic Control | norm_wave | $[0.7, 1.3]$ |
| | norm_wait | $[-1.2, -0.8]$ |
| | init_density | $[-0.3, 0.3]$ |
| | coop_gamma | $[0.5, 1.1]$ |
| | headway_min | $[0.5, 1.5]$ |
| | headway_st | $[4.5, 6.5]$ |
| | headway_go | $[30, 40]$ |
| | speed_max | $[25, 35]$ |
| | accel_max | $[2, 3]$ |
| | accel_min | $[-3, -2]$ |

- **quads_collision_falloff_radius**: The falloff radius determines how collision effects diminish with increasing distance between quadrotors, with a range of $[0.7, 1.3]$. Within this radius, the interference forces decrease gradually, affecting flight dynamics and reducing abrupt changes in quadrotor behavior during near collisions.

- **quads_obst_density**: This parameter, ranging from $[-0.2, 0.2]$, represents the relative density of obstacles in the environment. Positive values indicate a denser obstacle field, increasing the difficulty of navigation, while negative values suggest a sparser environment, facilitating smoother flight.

- **quads_obst_size**: The size of obstacles is represented by this parameter, with a range of $[0.7, 1.3]$. Larger obstacles necessitate more substantial avoidance maneuvers, whereas smaller obstacles, despite being easier to evade individually, may pose challenges when present in large numbers.

- **quads_collision_reward**: This parameter defines the reward value associated with collisions between quadrotors, with a range of $[-0.3, 0.3]$. Negative values penalize collisions, encouraging the control algorithm to prioritize collision avoidance. The magnitude of the reward/penalty influences the aggressiveness of the avoidance maneuvers.

- **quads_collision_smooth_max_penalty**: This parameter specifies the maximum penalty for non-smooth collision avoidance maneuvers, with a range of $[7, 13]$. It ensures that the quadrotors' movements remain smooth and controlled, even when avoiding collisions. Higher penalties encourage more gradual and stable maneuvers, reducing the likelihood of abrupt changes in velocity or direction.
  - **quads_episode_duration**: This parameter defines the duration of each control episode, ranging from $[10, 20]$ seconds. It determines the time horizon over which the control algorithm must manage the quadrotor swarm. Longer durations require more robust and sustained control strategies, while shorter durations allow for more aggressive maneuvers.
  - **quads_obst_collision_reward**: This parameter defines the reward value associated with collisions between quadrotors and obstacles, with a range of $[-0.3, 0.3]$. Negative values penalize such collisions, encouraging the control algorithm to prioritize obstacle avoidance. The magnitude of the reward/penalty influences the aggressiveness of the avoidance maneuvers.
  - **replay_buffer_sample_prob**: This parameter defines the probability of sampling from the replay buffer in reinforcement learning algorithms, with a range of $[0.5, 1]$. It affects how frequently the algorithm revisits past experiences stored in the replay buffer. Higher sampling probabilities ensure that the algorithm makes better use of historical data, improving its learning efficiency and ability to generalize from past experiences.

- **Active Voltage Control**:
  - **pv_scale**: This parameter adjusts the proportion of photovoltaic (PV) power generation in the grid, with values ranging from $[0.7, 1.3]$. A value above 1 indicates an increased PV capacity, potentially enhancing voltage support, whereas values below 1 suggest reduced PV contributions, requiring supplementary power sources.
  - **demand_scale**: The scaling factor for electricity demand ranges from $[0.7, 1.3]$. Higher values signify increased power demand, complicating voltage regulation, while lower values ease the control requirements due to reduced demand.
  - **v_upper**: This parameter sets the upper voltage limit, with a range of $[0.705, 1.305]$. Ensuring grid voltage stays below this threshold is critical to prevent overvoltage-related equipment damage and maintain grid stability.
  - **v_lower**: The lower voltage limit ranges from $[0.65, 1.25]$. Maintaining voltage above this threshold is essential to prevent equipment malfunctions and avoid voltage collapse, thus ensuring reliable grid performance.
  - **q_weight**: This parameter represents the weighting factor for reactive power control, with values ranging from $[0, 0.2]$. A higher weight emphasizes the importance of reactive power management in the control strategy, helping to maintain voltage stability through reactive power compensation.
  - **action_bias**: The bias term for control actions ranges from $[-0.5, 0.5]$. This parameter introduces a baseline offset in the control actions, which can be used to adjust the default behavior of the control algorithm. Positive values shift the control actions towards more active interventions, while negative values reduce the aggressiveness of the control actions.
  - **action_scale**: This parameter scales the magnitude of control actions, with values ranging from $[0.4, 1.2]$. A higher scale factor increases the impact of control actions on the system, allowing for more significant adjustments to voltage levels. Conversely, a lower scale factor results in more conservative control actions.
  - **demand_scale**: The scaling factor for electricity demand ranges from $[0.5, 1.5]$. This updated range indicates a broader variation in power demand, with higher values representing significantly increased demand scenarios that pose greater challenges for voltage regulation. Lower values, closer to 0.5, suggest reduced demand conditions that may simplify control requirements.

- **Dexterous Hand Manipulation**:
  - **startPositionNoise**: This parameter introduces initial position noise in dexterous hand manipulation tasks, ranging from $[-0.005, 0.005]$. It simulates minor deviations in the hand's starting position, increasing the complexity of the task and requiring precise control adjustments.

- **transition_scale**: The scaling factor for state transitions ranges from $[0.2, 0.8]$. Higher values result in more dynamic state changes, demanding robust control strategies, while lower values yield smoother transitions, facilitating easier manipulation.

- **orientation_scale**: This parameter, ranging from $[-0.2, 0.4]$, adjusts the amplitude of object orientation changes. Positive values enhance orientation variations, challenging control precision, while negative values dampen these changes.

- **rotRewardScale**: The rotation reward scaling factor, with a range of $[0.7, 1.3]$, influences the prioritization of rotational actions in control algorithms. Higher values incentivize exploring rotational strategies, potentially improving manipulation performance.

- **startRotationNoise**: This parameter introduces initial rotation noise in the hand's orientation, ranging from $[-0.5, 0.5]$. It simulates variability in the starting orientation of the hand, adding complexity to the task and requiring the control algorithm to adapt to different initial conditions.

- **resetPositionNoise**: This parameter specifies the noise added to the hand's position during resets, with values ranging from $[-0.3, 0.3]$. It introduces randomness in the hand's position at the beginning of each trial, making the task more challenging and requiring the control algorithm to handle a wider range of initial positions.

- **resetRotationNoise**: This parameter defines the noise added to the hand's rotation during resets, with values ranging from $[0.0, 0.01]$. It introduces small perturbations in the hand's orientation at the start of each trial, requiring the control algorithm to correct for these initial deviations.

- **resetDofPosRandomInterval**: This parameter specifies the random interval for resetting the degrees of freedom (DoF) positions, ranging from $[0, 0.4]$. It introduces variability in the initial joint positions of the hand, making the task more realistic and challenging, as the control algorithm must adapt to different starting configurations.

- **resetDofVelRandomInterval**: This parameter defines the random interval for resetting the degrees of freedom (DoF) velocities, with values ranging from $[-0.2, 0.2]$. It introduces variability in the initial joint velocities of the hand, adding an additional layer of complexity to the task and requiring the control algorithm to manage both position and velocity adjustments.

- **Intelligent Traffic Control**:

  - **norm_wave**: This parameter represents normalized traffic flow fluctuations, ranging from $[0.7, 1.3]$. Larger values indicate greater variability in traffic flow, increasing the complexity of traffic management, while lower values suggest more stable traffic patterns.

  - **norm_wait**: The normalized waiting time parameter ranges from $[-1.2, -0.8]$. Negative values denote a reduction in average vehicle waiting times, typically resulting from optimized control measures, thereby improving overall traffic efficiency.

  - **init_density**: This parameter, with a range of $[-0.3, 0.3]$, defines the initial traffic density. Positive values indicate congested conditions that require advanced control strategies, whereas negative values suggest lighter traffic, allowing simpler interventions.

  - **coop_gamma**: This parameter, ranging from $[0.5, 1.1]$, represents the cooperation factor in traffic control. Higher values indicate stronger cooperation among vehicles or traffic control systems, which can improve traffic flow efficiency. Lower values suggest more independent behavior, potentially leading to less coordinated traffic patterns.

  - **headway_min**: The minimum headway distance between vehicles ranges from $[0.5, 1.5]$ meters. This parameter defines the smallest safe distance that should be maintained between vehicles to avoid collisions. A smaller minimum headway allows for higher traffic density but increases the risk of accidents.

  - **headway_st**: The standard headway distance ranges from $[4.5, 6.5]$ meters. This parameter represents the typical distance maintained between vehicles under normal driving conditions. It affects the overall traffic flow and capacity of the road.

  - **headway_go**: The headway distance for vehicles in motion ranges from $[30, 40]$ meters. This parameter is particularly relevant for high-speed traffic and defines the

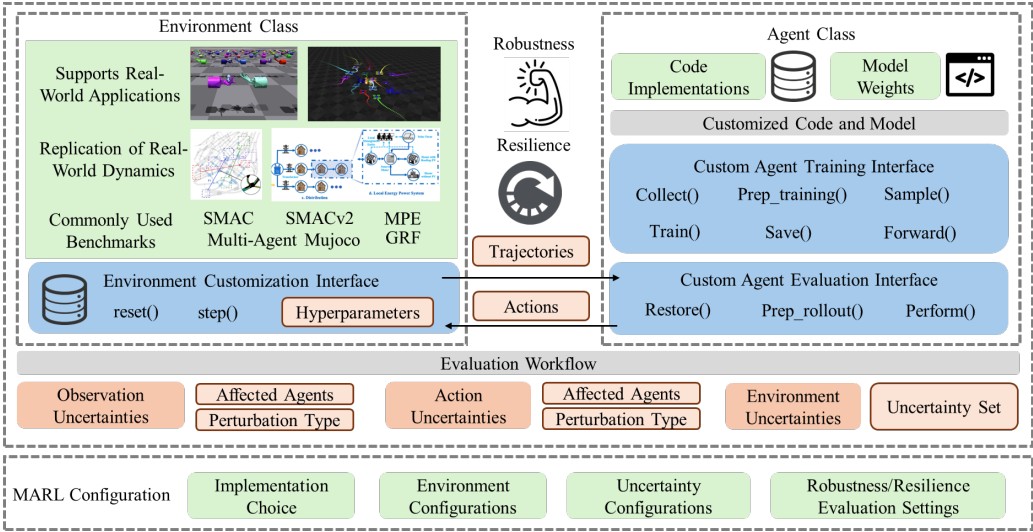

Figure 11: Architecture of our codebase. The framework supports customized environments and algorithms, provided they implement the required interfaces. The evaluation of hyperparameters on robustness and resilience is integrated into an automated workflow.

- safe distance required to maintain smooth traffic flow while allowing for safe stopping distances.

  – **speed_max**: The maximum allowable speed ranges from $[25, 35]$ meters per second (or approximately 90 to 126 kilometers per hour). This parameter sets the upper limit for vehicle speeds, balancing traffic flow efficiency with safety considerations.

  – **accel_max**: The maximum acceleration ranges from $[2, 3]$ meters per second squared. This parameter defines the highest rate at which vehicles can increase their speed, affecting traffic dynamics and the ability to manage traffic flow efficiently.

  – **accel_min**: The minimum acceleration (or maximum deceleration) ranges from $[-3, -2]$ meters per second squared. This parameter defines the highest rate at which vehicles can decelerate, which is crucial for maintaining safe distances and avoiding rear-end collisions.

These parameters are sampled randomly to reflect real-world variability. In MARL scenarios, as all agents are equally influenced by environmental changes, we denote this type of uncertainty as *env*.

## A.4  Additional Details on Architectures

In this section, we detail the overall architecture of our algorithm, as shown in Fig. 11. Our framework supports customized environments and algorithms, provided they implement the required interfaces. The evaluation of hyperparameters on robustness and resilience is integrated into an automated workflow. Our architecture design are also inspired by existing prominent codebase, including [3, 4, 79, 80, 81].

### A.4.1  Environment Architecture

To ensure the comprehensiveness and utility of our codebase, we integrate three types of environments to support diverse applications and research needs.

The first category includes environments that support real-world applications. For instance, in *Dexterous Hand Manipulation* [53], two agents collaboratively control a 24-DoF Shadow Hand [82] to perform complex tasks similar to those learned by infants. Similarly, in *Quadrotor Swarm Control* [54], multiple Crazyflie quadrotors [83] are trained to execute advanced swarm robotics tasks. Policies trained in these simulation environments can be directly deployed to real-world robots with minimal Sim2Real gaps.

The second category comprises environments constructed from real-world data to ensure high-fidelity replication of real-world dynamics. For example, in *Intelligent Traffic Control* [55], agents optimize adaptive traffic signal control (ATSC) using real-world traffic data from Monaco and perform cooperative adaptive cruise control (CACC), a critical task in autonomous driving. In *Active Voltage Control*, agents regulate voltage fluctuations to remain within IEEE grid standards, as described by [68, 69]. These environments provide high-fidelity simulations of real-world traffic systems and voltage control scenarios.

The third category includes widely used benchmark environments in MARL research. These include SMAC [84], SMAC v2 [85], MPE [1], Multi-Agent Mujoco [86], and Google Research Football [87]. We believe these benchmarks are representative of environments commonly used in MARL studies, offering researchers a solid foundation for their work on robust MARL.

Finally, recognizing the rapid emergence of new MARL environments and the existence of proprietary industrial environments, we designed our codebase to support customization. Our environment interface accommodates custom environments as long as they implement two core APIs: `reset()` and `step()`. The `reset()` function initializes a new episode, while `step()` advances the environment to the next timestep based on agent actions. To evaluate the robustness and resilience against environment uncertainties, our interface also take custom environment hyperparameters as inputs.

### A.4.2 Agent Architecture

As for the architectures of each MARL agent, we aim to make our design general and adaptable, accommodating customized code implementations and model weights. Our architecture is as follows:

To evaluate the robustness and resilience of MARL algorithms, we introduce an `Agent` class that disentangles code implementations and model weights from the evaluation process. This approach differs from previous codebases such as Epymarl [42] and MAPPO [3], which separate agent functionality into three classes: actor, critic, and mixer. Such a separation complicates integration, making the pipeline less user-friendly, particularly for non-MARL researchers.

In our codebase, we abstract all functionalities required for MARL agents into a single `Agent` class. This design allows users to integrate their custom code implementations and model weights into our framework by implementing APIs that satisfy our requirements. Specifically, the `Agent` class represents MARL agents interacting with the environment by receiving trajectories and outputting joint actions. If training is not required during evaluation, users need only upload their code implementation and model weights, and implement the following three APIs:

- `restore()`: Loads model weights into the code implementation.
- `prep_rollout()`: Initializes the decision-making agent, including clearing the hidden state of RNNs and resetting the action distribution.
- `perform()`: Outputs the agent's action distribution during testing.

If training with customized code implementations is required, users need to additionally implement the following six APIs:

- `collect()`: Collects trajectory data for all agents, including state, individual observations, and rewards.
- `prep_training()`: Initializes the decision-making agent for training, clearing RNN hidden states and resetting the action distribution. Note that some algorithms, such as MADDPG, use different action distributions for training and testing, which this API supports.
- `sample()`: Samples an action from the model output. This varies by algorithm (e.g., QMIX, MAPPO, MADDPG).
- `train()`: Trains the model using data collected from the environment. While training methods differ across algorithms, all training is encapsulated within this single API, simplifying integration.
- `save()`: Saves the trained model weights.
- `forward()`: Processes observation input and outputs an action distribution for all agents. For Q-learning algorithms, this API returns Q-values for each action and agent.

Although similar APIs exist in previous codebases, our work is the first to propose abstracting an `Agent` class to enable seamless integration of customized implementations and model weights. This abstraction simplifies previous approaches [42, 3], which required the separate implementation of actor, critic, and mixer classes. By consolidating functionalities, our framework enhances usability and flexibility for both researchers and practitioners.

### A.4.3 Uncertainty Architecture

Next, we describe the architecture for applying uncertainties to MARL systems. The primary design principle is to ensure that uncertainties are simple to implement and operate in parallel with the existing `Environment` and `Agent` classes, without interfering with their original functionality.

Uncertainties are introduced during the interaction between the `Environment` and `Agent` classes. As illustrated in Fig. 11, the three types of uncertainties are highlighted in orange and applied as follows:

- **Observation uncertainty**: This is added to the observation inputs of all agents. The perturbations are applied after the agents collect trajectories from the environment.
- **Action uncertainty**: Actions, generated by the `Agent` class and passed to the `Environment` class, are perturbed after the victim agent outputs its action probabilities.
- **Environment uncertainty**: These uncertainties are sampled from predefined external uncertainty sets and applied by modifying the environment's hyperparameters.

By structuring the addition of uncertainties in this manner, we achieve a concise and modular implementation that integrates seamlessly with the existing architecture.

### A.4.4 Efficient Evaluation Workflow

Given the extensive range of uncertainties, hyperparameters, tasks, and algorithms integrated into our codebase, an efficient evaluation workflow is essential. Our system allows users to specify the desired uncertainties, hyperparameters, tasks, and algorithms for evaluation. Once configured, the workflow automatically generates a comprehensive set of shell commands, facilitating large-scale, automated evaluations. This significantly reduces the time and effort required by users, streamlining the evaluation process and enabling rapid experimentation at scale.

For example, to train and evaluate all hyperparameters under different uncertainties, one sample code for generating all bash commands is:

Listing 1: Code for generating evaluation workflow.

```
python generate.py eval -e dexhands -s ShadowHandSwingCup -a mappo --
    ↪ extra="--headless" --stage 1
```

This automatically generates 256 commands for execution.

Listing 2: Generated bash commands for training and evalution.

```
CUDA_VISIBLE_DEVICES=0 python -u /root/adv_marl_benchmark/
    ↪ single_train.py --algo mappo --env dexhands --env.task
    ↪ ShadowHandSwingCup --algo.use_eval True --algo.log_dir ./
    ↪ results --algo.slice True --load_victim ./results/dexhands/
    ↪ ShadowHandSwingCup/single/mappo/default/seed
    ↪ -00002-2024-11-24-00-23-30 --exp_name random_noise_default --
    ↪ run perturbation --algo.num_env_steps 0 --algo.perturb_iters 0
    ↪  --algo.adaptive_alpha False --algo.targeted_attack False --
    ↪ headless > out/logs/dexhands/ShadowHandSwingCup/mappo/default/
    ↪ random_noise.log 2>&1
CUDA_VISIBLE_DEVICES=0 python -u /root/adv_marl_benchmark/
    ↪ single_train.py --algo mappo --env dexhands --env.task
    ↪ ShadowHandSwingCup --algo.use_eval True --algo.log_dir ./
    ↪ results --algo.slice False --load_victim ./results/dexhands/
    ↪ ShadowHandSwingCup/single/mappo/activation_func_leaky_relu/
    ↪ seed-00002-2024-11-25-05-51-49 --exp_name
    ↪ random_noise_activation_func_leaky_relu --run perturbation --
```

```
        ↪ algo.num_env_steps 0 --algo.perturb_iters 0 --algo.
        ↪ adaptive_alpha False --algo.targeted_attack False --headless >
        ↪  out/logs/dexhands/ShadowHandSwingCup/mappo/
        ↪ activation_func_leaky_relu/random_noise.log 2>&1
3  ...
4  CUDA_VISIBLE_DEVICES=0 python -u /root/adv_marl_benchmark/
        ↪ single_train.py --algo mappo --env dexhands --env.task
        ↪ ShadowHandSwingCup --algo.use_eval False --algo.log_dir ./
        ↪ results --algo.slice False --load_victim ./results/dexhands/
        ↪ ShadowHandSwingCup/single/mappo/use_recurrent_policy_True/seed
        ↪ -00002-2024-11-26-01-18-24 --exp_name
        ↪ pert_obs_sin_use_recurrent_policy_True --run perturbation --
        ↪ algo.num_env_steps 0 --algo.perturb_iters 10 --algo.
        ↪ adaptive_alpha True --algo.targeted_attack False --adv_all
        ↪ False --adv_eps 0.1 --headless > out/logs/dexhands/
        ↪ ShadowHandSwingCup/mappo/use_recurrent_policy_True/
        ↪ pert_obs_sin.log 2>&1
```

# B   Additional Details on Experiments and Takeaways

## B.1   Details of Reward Normalization Methods

In reinforcement learning, the scale of episodic rewards can vary substantially across tasks. This variability complicates the consistent evaluation of RL algorithms under different experimental settings. To enable robust and systematic comparisons, we adopt three reward normalization methods. These approaches ensure meaningful performance assessments across diverse uncertainties and hyperparameters.

- **Z-Score Normalization**: To address task-specific variations in episodic rewards ($R$), we employ z-score normalization on a per-task basis. Each reward value is transformed by subtracting the mean reward for its respective task and dividing by the standard deviation of the task's rewards. The normalized reward is defined as:

$$R_{\text{norm}} = \frac{R - \mu_{\text{task}}}{\sigma_{\text{task}}}$$

  where $R$ is the original reward, $\mu_{\text{task}}$ denotes the mean reward for the task, and $\sigma_{\text{task}}$ is the standard deviation of the task's rewards. These statistics are calculated over the full reward distribution for the task, encompassing rewards from both untrained models and models trained with all algorithms under all uncertainties and hyperparameters in the benchmark. In our analysis, unless otherwise specified, z-score normalization is applied wherever rewards from different tasks are aggregated into a single distribution, such as in the correlation analysis presented in Section 5.1 and the uncertainty diversity analysis in Section 5.2.

- **Performance Normalization for hyperparameters**: To examine the impact of hyperparameters on each task (as discussed in Section B.4), we rescale the rewards to quantify the variance caused by different hyperparameters. Specifically, for each task, we normalize the rewards using the formula:

$$R_{\text{norm}} = \frac{R_{\text{impl}} - R_{\text{untrain}}}{R_{\text{default}} - R_{\text{untrain}}}$$

  where $R_{\text{impl}}$ is the episodic reward obtained under a specific hyperparameters, $R_{\text{untrain}}$ is the reward of the untrained model, and $R_{\text{default}}$ is the reward achieved using the default hyperparameters for the task.

  After normalizing the rewards, we aggregate the rewards across all algorithms, uncertainties, and hyperparameters for each task. The resulting reward distribution is then averaged over the dimensions of algorithms and uncertainties, ensuring that the remaining variance in the reward distribution is solely attributed to differences in hyperparameters. Then, the impact of hyperparameters can be quantified by computing the 95% confidence interval of the normalized reward distribution for each task.

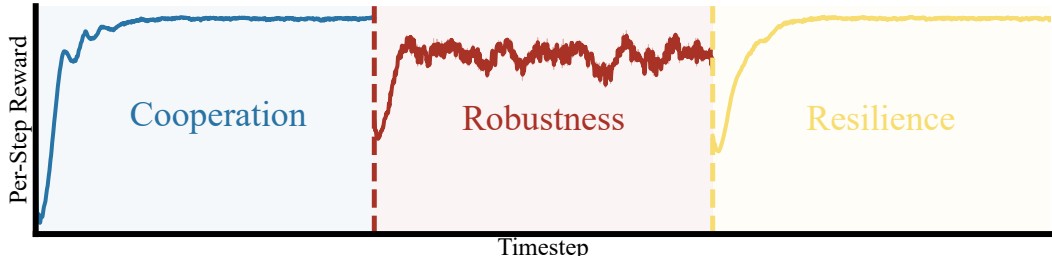

(a) Evaluation curve for cooperation, robustness and resilience.

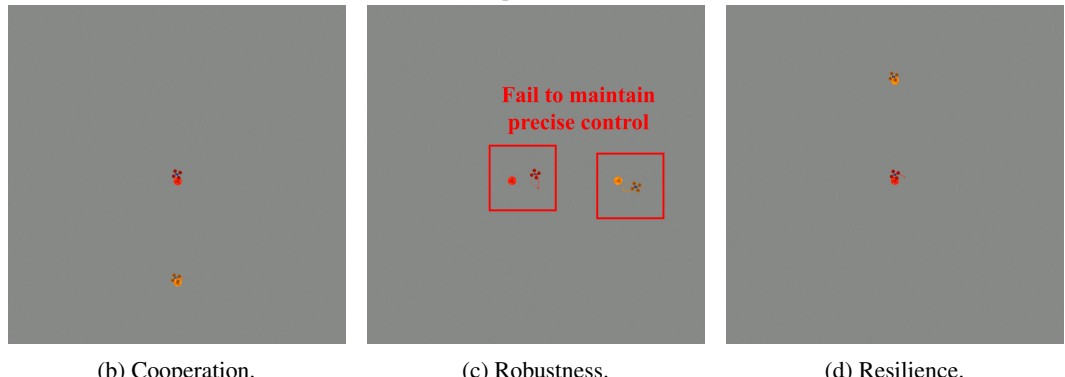

(b) Cooperation.       (c) Robustness.       (d) Resilience.

Figure 12: Case study on agents that are robust but non-resilient. Evaluated in task **static_diff_goal** in Quads environment, under the uncertainty setting **obs_greedy_all**.

- **Relative Performance Change by hyperparameters**: To further examine the efficacy of specific hyperparameters (see Section 5.3), we calculate the relative performance change of each hyperparameters with respect to the default implementation. This metric captures the degree to which an hyperparameters improves or degrades task performance. The normalized reward is defined as:

$$R_{\text{norm}} = \frac{R_{\text{impl}} - R_{\text{default}}}{R_{\text{default}} - R_{\text{untrain}}}$$

In this equation, $R_{\text{impl}}$ is the reward achieved with a specific hyperparameters, $R_{\text{default}}$ is the reward obtained using the default implementation, and $R_{\text{untrain}}$ is the reward of the untrained model.

By explicitly quantifying relative performance changes, this method highlights hyperparameters that significantly enhance or impair task performance. It offers a clear and interpretable metric for evaluating the effectiveness of various implementation strategies under different experimental conditions.

## B.2 Examples on Differences Between Robustness and Resilience

In this section, we present two representative cases to illustrate the distinction between robustness and resilience. Specifically, we provide examples of agents that are robust but not resilient, and agents that are resilient but not robust. To ensure visual clarity, we restrict our analysis to the Quads environment. Other environments, such as Traffic and Voltage, are excluded due to limitations in rendering quality, while the fine-grained control of each finger joint required in Dexhands is difficult to interpret intuitively.

**Robust but non-resilient.** We examine a case of robustness without resilience in the task **static_diff_goal** within the Quads environment, under the uncertainty setting **obs_greedy_all**. This task involves two quadrotors, each assigned a distinct goal state. The objective is to navigate each quadrotor to its respective goal. The uncertainty **obs_greedy_all** introduces small observation noise to all agents, perturbing their perceptions toward adversarial directions, as described in [30].

As shown in Fig. 12a, the system performs comparably well under cooperative and resilient conditions, yet fails under robustness evaluations. In particular, in Fig. 12b and Fig. 12d, agents behave similarly

when initialized in cooperative configurations or when recovering post-attack, with both quadrotors successfully navigating to their respective goals. However, under observation noise, as shown in Fig. 12c the agents are still able to reach approximate goal regions but **fail to maintain precise control**. The continuous perturbations in their observations lead to control errors that gradually displace the quadrotors from their true goals. This inability to maintain fine control reflects a lack of robustness. In contrast, a truly robust agent would complete the task accurately despite the presence of such uncertainties—an essential requirement for real-world deployment.

**Resilient but non-robust.** We now consider a case that exhibits resilience but lacks robustness, under the task **o_static_same_goal** in the Quads environment, with uncertainty **env**. This task features two quadrotors, multiple obstacles, and a shared goal. The uncertainty **env** modifies the environment dynamics by increasing the number and size of obstacles.

As shown in Fig. 13b and 13c, the policy enables the agents to reach the goal even in the presence of added obstacles, albeit with minor collisions. This suggests a degree of robustness to environment-level perturbations. However, when evaluating resilience in isolation (*i.e.*, in the absence of uncertainty), as shown in Fig. 13d the quadrotors often take suboptimal trajectories, drifting away from the goal and failing to reach it precisely. We attribute this failure to a **mismatch in state initialization**: during resilience evaluation, both quadrotors begin in close proximity to the goal (Fig. 13d 1)), whereas in cooperative and robustness evaluations, they are initialized farther away (Fig. 13b 1) and 13c 1)). This discrepancy leads to differing dynamics during execution.

In cooperative and robustness settings, the greater distance from the goal encourages smooth acceleration and stable convergence. In contrast, during resilience evaluation, the short initial distance prompts one quadrotor to accelerate too quickly, overshooting the target and needing to reverse course (Fig. 13d 2)). Upon returning, both quadrotors struggle to stabilize at the goal—an issue not observed under other evaluation conditions. This behavior consistently emerges across multiple rollouts. We attribute this to *state aliasing*: the agents encounter observation histories that closely resemble those seen during training, but differ in nuanced ways that demand distinct responses. The policy, unable to disambiguate these subtle variations, fails to adapt its behavior accordingly—resulting in degraded fine-grained control and impaired resilience performance (Fig. 13d 3)).

## B.3 Additional results on generality of linearity and further discussions

In this section, we discuss the linearity of correlation of performance degradation with robustness and resilience, under different uncertainty types, agent scopes, and attack strategies. As shown in Fig. 14, while linearity does not hold universally, it is statistically significant ($p < .05$) in most cases—including random, greedy, and optimal attacks, as well as observation uncertainty and perturbations applied to single or all agents. For environment uncertainty, linearity holds ($r = 0.99$ for robustness, $r = 0.85$ for resilience) but lacks statistical significance. Nonetheless, cooperation strongly correlates with both robustness and resilience under environment uncertainty ($r > 0.88$ and $r > 0.95, p < .001$), suggesting that improving cooperation enhances robustness in this modality. In contrast, action uncertainty shows weak linearity (e.g., $r = 0.37, p = 0.19$ for robustness), largely due to MADDPG's high resilience to single-agent perturbations. When these outliers are removed, linearity improves significantly ($r = 0.69, p < .05$ for robustness; $r = 0.91, p < .001$ for resilience), reflecting MADDPG's off-policy training and robustness to exploratory noise.

## B.4 How Important Are hyperparameters?

In this section, we highlight the importance of hyperparameters in MARL by analyzing their impact across tasks. For each task, we quantify this impact as the 95% confidence interval of relative performance changes due to hyperparameters (See details in Appendix B.1). Our key findings are:

**1. hyperparameters affect tasks unevenly.** As shown in Fig. 15, the impact of hyperparameters varies widely. Cooperative performance can change by 190%, robustness by 1123%, and resilience by 1058% with a single change in hyperparameters. This underscores the need of practitioners to identify and focus on tasks highly sensitive to such changes.

**2. hyperparameters often outweigh algorithms.** In Fig. 15, we highlight tasks in *blue* where a two-way ANOVA shows that changing a single hyperparameters has a more significant effect than

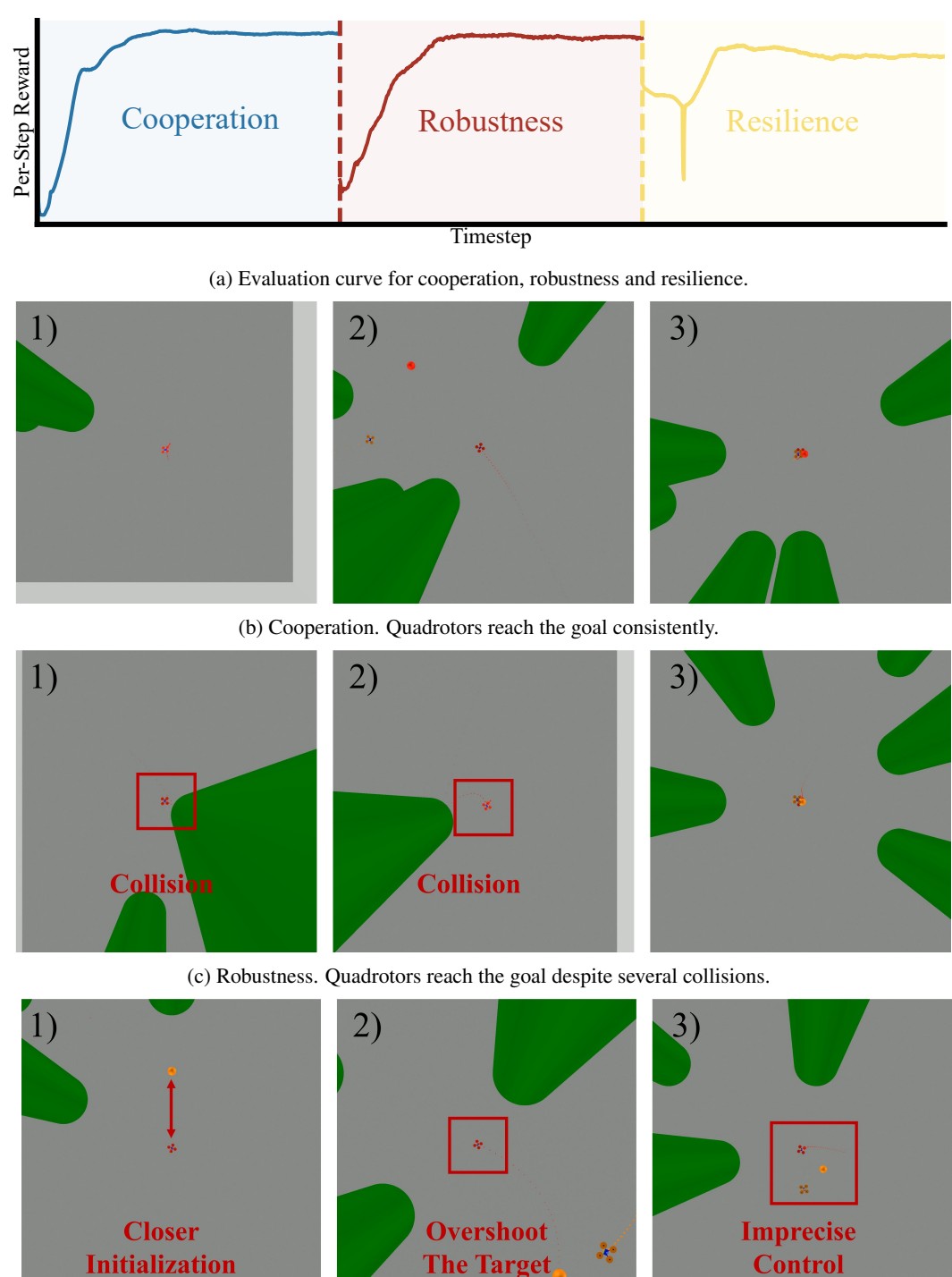

(a) Evaluation curve for cooperation, robustness and resilience.

(b) Cooperation. Quadrotors reach the goal consistently.

(c) Robustness. Quadrotors reach the goal despite several collisions.

(d) Resilience. Quadrotors are initialized in an unseen state closer to the goal, making a fast acceleration and overshooting the target. This overshooting brings state aliasing in subsequent histories, resulting in imprecise control.

Figure 13: Case study on agents that are Resilient but non-robust. Evaluated in task **o_static_same_goal** in Quads environment, under the uncertainty setting **env**.

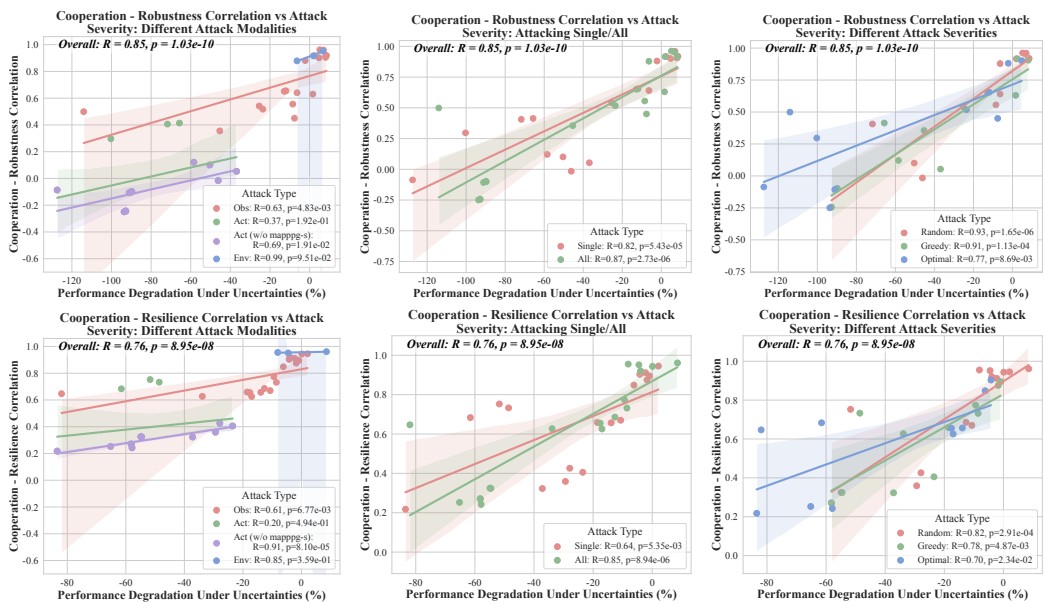

Figure 14: Linearity of robustness and resilience correlations under different uncertainty types, agent scopes, and attack strategies.

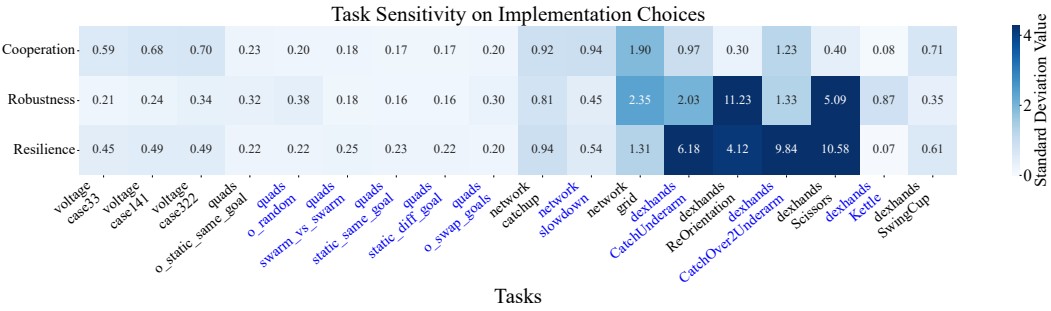

Figure 15: The impact of hyperparameters varies across different tasks. Task names in *blue* indicate cases where altering a single hyperparameters has a greater effect on performance than changing the MARL algorithm itself (two-way ANOVA, $p < .001$).

Table 6: Examining the effect on the use of parameter sharing, GAE and PopArt on commonly used benchmarks, including MPE, SMAC and Multi-Agent Mujoco (MA-Mujoco). In fact, while these tricks are effective on SMAC, the performance gain is not consistent across all environments.

| Environment | Task | Default | Use_GAE_False | Use_PopArt_False | Share_Param_False |
|---|---|---|---|---|---|
| MPE | Speaker-Listener | $-11.48$ | $-11.39$ | $-11.39$ | $-11.40$ |
| | Spread | $-27.56$ | $-26.12$ | $-31.86$ | $-27.25$ |
| SMAC | 3m | 18.62 | 18.80 | 18.75 | 11.58 |
| | 2s3z | 28.74 | 28.07 | 28.80 | 25.21 |
| MA-Mujoco | Ant-4x2 | 12256.34 | 8252.60 | 8715.61 | 17260.66 |
| | HalfCheetah-6x1 | 19272.95 | 6628.80 | 8876.22 | 28280.03 |

switching algorithms ($p < .001$). Remarkably, in 9 out of 18 tasks, the performance improvement achieved by altering a single hyperparameters surpasses that of switching between MARL algorithms.

**Takeaway.** Practitioners should evaluate task sensitivity to hyperparameters and prioritize tasks with high variability, as hyperparameters can have a profound impact on MARL performance.

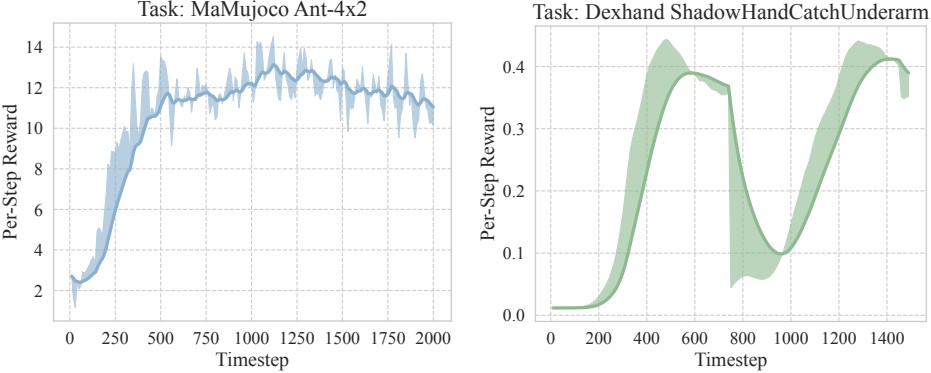

Figure 16: Verifying our hypothesis that GAE is ineffective. GAE works best in Multi-Agent Mujoco with dense and predictable reward, while in Dexhand, the reward is less predictable.

## B.5 Evaluating the Effectiveness of hyperparameters on Other Environments

To verify the counterintuitive conclusion that the use of parameter sharing, GAE and PopArt is not a generally effective hyperparameters, we check the performance of these tasks in three commonly used benchmarks, MPE, SMAC and Multi-agent Mujoco using MAPPO. We select two task for each environment including speaker_listener and spread for MPE, 3m and 2s3z for MAPPO, and Ant-4x2, HalfCheetah6x1 for Multi-Agent Mujoco. We report the results in 5 random seeds.

As shown in Table 6, we confirm MAPPO's finding that parameter sharing is critical for SMAC tasks. However, parameter sharing offers no benefit in Multi-Agent Mujoco, where the task involves controlling highly diverse joints, and its impact is mild in MPE. Additionally, while GAE and PopArt significantly improve performance in Multi-Agent Mujoco, they have negligible or even slightly negative effects in MPE and SMAC. These findings support the statements in our main paper: (1) parameter sharing is effective for SMAC and not universally beneficial, and (2) the effectiveness of GAE and PopArt is task-dependent and not universally applicable.

## B.6 Verifying the hypothesis of the ineffectiveness of GAE

In this section, we conduct a preliminary analysis to test our hypothesis that GAE improves performance in Multi-Agent MuJoCo but is less effective for real-world tasks. As shown in Fig. 16, the rewards in Multi-Agent MuJoCo are dense, stable, and predictable within an episode. Consequently, GAE effectively stabilizes learning and reduces variance. In contrast, Dexhand presents a more complex task with highly variable and less predictable rewards over time. As a result, GAE may introduce significant errors in estimating $\delta$, leading to larger bootstrapping errors during value function approximation.

## B.7 Analyzing the synergy of hyperparameters

In this section, we analyze how individual hyperparameters interact when jointly applied, as shown in Fig. 8, and identify those that remain consistently beneficial in combination.

To quantify their joint effect, we leverage the individual impact of each hyperparameter from Fig. 6 (denoted as $h_i$) and the combined performance when applying the best group of hyperparameters from Fig. 8 (denoted as $j$). We fit an ordinary least squares (OLS) regression model, assigning a weight $w_i$ to each individual hyperparameter $v_i$, and minimize the squared error $(j - \sum_i h_i w_i - \epsilon_i)^2$, where $\epsilon_i$ is a random noise. While OLS shares similar formulation of standard linear regression, it additionally enables significance testing by adding random noise, allowing us to filter out spurious correlations and identify hyperparameters that offer consistent synergistic effects.

We apply this OLS analysis separately for cooperation, robustness, resilience, and the combined dataset, using 5% winsorization to suppress outliers. A higher weight $w_i$ indicates a stronger contribution of the corresponding hyperparameter when used in combination. Note that the learned weights do not reflect global average performance, but rather the effectiveness of each hyperparameter variation within its specific task. As a result, some variations may not perform well on average but

Table 7: OLS Regression Coefficients for Each Hyperparameter Across Metrics, with statistical test results denoted by $*$. $*$ denotes $p < .05$, $**$ denotes $p < .001$.

| Hyperparameter | Cooperation | Robustness | Resilience | All metrics |
|---|---|---|---|---|
| use_popart_False | 0.6688 | 1.6617$^{**}$ | 1.1007 | 2.0496$^{**}$ |
| gamma_0.95 | 2.8845$^{**}$ | 1.1021$^{**}$ | 1.0425$^{**}$ | 1.5002$^{**}$ |
| gamma_0.9 | 2.7529$^{**}$ | 0.5963$^{**}$ | 1.0765$^{**}$ | 1.3207$^{**}$ |
| activation_func_selu | 3.7292$^{**}$ | 0.2915 | 1.2928$^{**}$ | 1.0862$^{**}$ |
| share_param_True | 0.8536$^{*}$ | 0.8999$^{**}$ | 0.7935$^{*}$ | 0.8932$^{**}$ |
| hidden_sizes_64_64 | 1.3729$^{**}$ | 0.7576$^{**}$ | 0.1808 | 0.8097$^{**}$ |
| critic_lr_0.005 | 0.9467$^{**}$ | 0.4166$^{*}$ | 0.8912$^{**}$ | 0.7751$^{**}$ |
| hidden_sizes_256_256 | $-0.6873$ | 0.9917$^{**}$ | 0.5153$^{*}$ | 0.7496$^{**}$ |
| initialization_method_xavier_uniform | $-1.1196^{*}$ | 0.6602$^{*}$ | 0.5090 | 0.7130$^{**}$ |
| entropy_coef_0.001 | 2.0065$^{**}$ | 0.6207$^{*}$ | 0.2690 | 0.6972$^{**}$ |
| n_step_5 | $-1.5451^{*}$ | 0.3846 | 0.3337 | 0.5887 |
| expl_noise_0.01 | $-3.1952$ | 0.7608 | $-1.3955$ | 0.5136 |
| entropy_coef_0.0001 | 0.3350 | 0.9233$^{*}$ | 0.1412 | 0.4918$^{*}$ |
| use_feature_normalization_False | $-0.4857^{*}$ | 0.4671 | 0.4711$^{*}$ | 0.3944$^{*}$ |
| use_recurrent_policy_True | $-0.7096^{*}$ | 0.5041$^{*}$ | $-0.2259$ | 0.0402 |
| expl_noise_1.0 | 0.0000 | 0.0000 | 0.0000 | 0.0000 |
| activation_func_sigmoid | $-1.2446^{**}$ | 0.7742$^{*}$ | $-0.0398$ | $-0.0698$ |
| n_step_25 | $-4.0924^{**}$ | 0.2335 | 0.2881 | $-0.0762$ |
| activation_func_tanh | $-0.4447$ | $-0.1760$ | $-0.1597$ | $-0.2164$ |
| activation_func_leaky_relu | $-0.9681^{*}$ | $-0.0915$ | $-0.1567$ | $-0.2396^{*}$ |
| expl_noise_0.001 | 0.7439 | $-0.1964$ | $-1.1622^{*}$ | $-0.3811$ |
| use_gae_False | $-0.2794$ | $-0.1803$ | $-0.7281^{**}$ | $-0.5672^{**}$ |
| entropy_coef_0.1 | $-1.1006^{**}$ | 1.2337$^{**}$ | 0.1215 | $-0.6574^{*}$ |
| expl_noise_0.5 | $-0.7096$ | $-0.2485$ | $-1.7288$ | $-0.7314$ |
| lr_5e-05 | $-0.9459^{*}$ | 0.4082 | $-0.1224$ | $-0.7915^{*}$ |
| lr_0.005 | $-2.9178^{**}$ | $-0.3520$ | $-0.7945^{**}$ | $-0.9257^{**}$ |
| critic_lr_5e-05 | $-1.6991^{**}$ | $-0.7381^{**}$ | $-0.9523^{**}$ | $-0.9954^{**}$ |

can be highly beneficial in particular environments. As shown in Table. 7, results are sorted by the weights learned from the combined dataset. The findings are as follows.

**1. share_param_true is effective for homogeneous agents.** In our earlier analysis, parameter sharing, as a hyperparameter originally emphasized by MAPPO as their core trick, appeared ineffective when applied in isolation across all tasks. However, it proves beneficial in the Traffic environment, where agents are homogeneous and share similar observations and objectives. Since we use the best-performing hyperparameter set per task, parameter sharing consistently yields strong results in this setting ($p < .05$). In contrast, in environments with heterogeneous agents, such as Dexhand, Quads, and Voltage, agents have specialized roles, making parameter sharing less effective.

**2. Small discount factor $\gamma$ benefits tasks with short episodes.** Although smaller discount factors show mixed results when applied individually, we find they perform well in combination with other hyperparameters ($p < .001$ for both $\gamma = 0.9$ and $\gamma = 0.95$). Further analysis reveals that they are particularly effective in tasks like Dexhand and Voltage, which have shorter episode lengths (80 and 200 steps, respectively). In such settings, lower $\gamma$ downweights distant rewards and promotes faster convergence. In contrast, environments with longer episodes—such as Traffic (1000 steps) and Quads (1600 steps)—benefit from higher $\gamma$ values to better account for long-term dependencies.

**3. Higher critic learning rate brings consistent benefit.** While beneficial as a standalone trick, using a higher learning rate for the critic also consistently improves performance when combined with other variations ($p < .05$). This result further supports the effectiveness of the two-timescale actor-critic framework, where a faster-updating critic stabilizes training for a slower-updating actor.

**4. The benefit of early stop remains consistent in hyperparameter synergy.** Although early stop is effective as a standalone technique, we find it remains consistently beneficial when combined with other hyperparameters. Notably, we exclude it from Table 7, as we save both the final model and the best-performing checkpoint, allowing us to directly assess its impact. On average, early stop improves cooperation by 2.85%, robustness by 13.62%, and resilience by 2.48%, with the improvement statistically significant (paired $t$-test, $p < .001$). Its effectiveness is intuitive—early stopping selects the best-performing model, making it generally applicable hyperparameter for trustworthy MARL.

**Takeaway.** Use parameter sharing for homogeneous agents, smaller discount factors for short episodes. Always use higher critic LR and early stop.

