# OpenReview forum: "Empirical Study on Robustness and Resilience in Cooperative Multi-Agent Reinforcement Learning"
_NeurIPS.cc/2025/Conference — NeurIPS 2025 poster_

### Official Review · Reviewer_PfFs · 2025-06-27

**Clarity:** 3
**Significance:** 2
**Originality:** 2
**Rating:** 4
**Confidence:** 4

**Summary:**

This paper presents an extensive empirical study to motivate the difference between robustness and resilience in cooperative multi-agent reinforcement learning.

From the introduction, both robustness and resilience are defined as the ability to cope with uncertainties.
The difference lies in the level of uncertainty faced by the agents, as the need for robustness becomes a need for resilience when uncertainty exceeds a certain threshold.
A first claim is that MARL literature uses both terms interchangeably, while control theory distinguishes between them.
The goal of the empirical study is to understand how hyperparameter tuning can address these two situations when training agents solely in their cooperative task without uncertainty.
The contributions are:
1: Cooperative MARL training leads to robustness and resilience if the uncertainty is small.
2: Being robust to one type of uncertainty does not imply being robust to another (e.g., uncertainty in obs vs in action)
3: Tuning the hyperparameter can improve robustness and resilience.

The classical Dec-POMDP is defined as a framework for performing cooperative tasks within an infinite horizon.
Three objectives are then defined:
Cooperation objective: classical expected sum of discounted rewards from an initial state with an infinite horizon
Robustness objective: Cooperative objective under uncertainty in obs, action, or environment.
Resilience objective: Cooperative without robustness' uncertainty, but with different initial state distributions.
It is implicit that Dec-POMDP components change when the evaluated objective is changed.
Agents are trained to maximise cooperation and are evaluated within Robustness and Resilience configurations.

The experimental procedure is thoroughly defined, either in the paper or in the appendix for additional information, including algorithms (3 popular CTDE algorithms), values of hyperparameters studied, real-world environments tested, types of uncertainties tested, and a complete code base is released.

Results analysed offer conclusions:
Training for cooperation can deal with small uncertainty.
Algorithms are distinguished based on their robustness and resilience against different types of uncertainty.
Dealing with one uncertainty does not imply the algorithm can deal with different ones.
Some hyperparameters are better to tune than others, and sometimes it is not the ones that are claimed in the literature to be the most effective.
Hyperparameter tuning can improve all three objectives at once.

Related work is finally presented, focusing on robustness and hyperparameter tuning in RL/MARL.
Some resilience-related works are mentioned with the definition of the resilience objective but not in this section.

The conclusion summarizes the claims and findings.

**Questions:**

1. From the introduction and Figure 2, resilience seems to evaluate the capacity of agents to recover from an unknown state after many noisy transitions, leading agents in unknown, or less explored, states.
Don't you think it is counterintuitive to define resilience as agents evolving in an environment with a new initial state distribution?
I guess that it makes sense formally, but it does not help the intuition behind.
Maybe you should emphasize that cooperation, robustness, and resilience configurations can be thought of as new episodes and not a single evolving one.
What about finite-time horizons?

2. Aren't resilience and generalisation the same?
Aside from RL and control, in machine learning, generalization refers to the model's ability to handle new data.
This seems to be really close, if not the same, as the resilience definition that is to have a good policy in new states.
This can also be said from robustness, tho, with more nuance.
I feel a statement on generalisation vs resilience can be interesting.
Especially when considering other methods than hyperparameter tuning to deal with it. I am thinking about regularisation from classical ML or all the methods in sim2real works.

3. About the hyperparameter. It is always difficult to choose them and their values. Maybe you can comment on the default choices and why you selected them. How did you choose the other values? Why not change the number of layers instead of hidden sizes? In TD-step, why 5 and 25 to compare with 20? Why not 5-20-35 or 15-20-25?

4. How would you define/compute the threshold in uncertainty when robustness becomes resilience?

5. It would be beneficial to include insights on how MADDPG and MAPPO/HAPPO behave differently facing different types of uncertainty.
Don't you think that simply the fact that the former has a discrete policy while others have a stochastic policy is the answer?

3. Why not train agents in robust and resilient configurations to investigate whether agents can learn them directly?
This could also improve the claim that being robust for one type of uncertainty does not imply being robust to all, and to improve all claims anyway.

**Ethical Concerns:**

["NO or VERY MINOR ethics concerns only"]

**Final Justification:**

The quality of the study is higher than the theoretical justifications.

**Limitations:**

In an empirical study, it is always possible to test more parameters/configurations, but I don't think it is a limitation in this paper.

To me, there is one limitation in the definition of resilience, how it relates to generalisation and the ambiguity between the uncertainty threshold and the new initial state distribution.

The main limitation is that hyperparameter tuning should not be the single or best option. I feel this is the message of the paper. There are many ways we can modify simulators or methods to address uncertainty. While some works addressing uncertainties with RL are clearly identified, the message that hyperparameter tuning on training agents for the task without uncertainty is the single solution to deal with uncertainty should be nuanced. It could be interesting to compare hyperparameter tuning for robustness without uncertainty at training against training agents with default hyperparameters, but in an environment with uncertainties.

**Quality:**

3

**Strengths And Weaknesses:**

In general, I would say that the quality of the study is higher than the impact of the claims.

Quality:

(+) The methods and parameters studied are well aligned with the state of the art.

(+) The empirical study and the details provided are of high quality.

(+) Technically, everything makes a lot of sense.

(-) A lot of math is defined to illustrate the study, but perhaps some developments, or an inequality between the three objectives, could improve the understanding.

Clarity:

(+) The paper is well written, and everything is clear.

(+) There is a huge amount of information in the appendix, making the paper "standalone", which is really appreciated.

(-) Robustness and resilience are first distinguished by a threshold of uncertainty. Afterwards, they are mathematically defined as different perturbations (noise on spaces vs noise on initial state distribution). The implicit idea is that agents tend to end up in unseen states due to high uncertainties, but there are ambiguities between these two definitions. The difference between resilience and robustness (lines 111-119) could be made earlier. Appendix B2 helps a lot (there is a mismatch between some paragraphs in B2, non-robust but resilient is the first one described) (suggestion: I feel instead of repeating lines 100-103, you can save some space to explain this)

(suggestion: I think it would be nice to group all the related works (l 103-107) in Section 5?

Significance:

(+) Such a study is really needed in the community, it seems the first of its kind in cooperative MARL.

(+) Using real-world environments for the study is a plus!

(+) The study advances the field towards deploying MARL.

(-) Hyperparameter tuning is presented as a single approach to address these problems, although others exist.

Originality:

(+) This study provides an additional push toward deploying RL in real-world applications.

(+) It is original to classify the effect of uncertainty based on the threshold.

(-) Resilience and generalisation seem to be highly related, but the latter is not mentioned.

---

> ### Author Rebuttal · Authors · 2025-07-30
>
> We thank Reviewer PfFs for appreciating the high quality, experiment design, writing and significance of findings of our paper. A detailed response is provided below.
>
> > Q1: Too many math included. Some developments, or an inequality between the three objectives, could improve the understanding.
>
> We will increase the clarity of these concepts by using power grid as an example:
>
> Cooperation describes optimal behavior under nominal conditions. For instance, on a typical day, the power grid functions in a coordinated and efficient manner, supplying energy smoothly across all nodes. This corresponds to high expected reward under ideal settings.
>
> Robustness refers to the ability to maintain performance during a disturbance. Suppose an earthquake causes sudden shifts in energy demand or damages part of the grid. A robust power grid can tolerate such perturbations, maintaining basic functionality despite degraded performance. This corresponds to a drop in reward, but without total failure.
>
> Resilience captures the system’s ability to recover after the disturbance. Once the earthquake ends and demand returns to normal, the grid must recover from a perturbed and potentially degraded state. A resilient system gradually restores optimal performance, resulting in increasing reward as it recovers.
>
> Regarding the equations, we acknowledge that too much math can hinder readability. However, since resilience has not been formally defined in prior literature, we believe the mathematical clarity is necessary, but will additionally add the example above to increase readability, such that readers can understand the concepts without relying solely on equations.
>
>
> > Q2: Ambiguities between these two definitions of robustness and resilience. The difference between resilience and robustness (lines 111-119) could be made earlier, group all the related works (l 103-107) in Section 5?.
>
> Thanks for your helpful suggestion. To reduce ambiguity, we will state the difference between resilience and robustness at the beginning of Section 2, and group all related works (Line 103-107) in Section 5 as suggested.
>
> > Q3: Resilience and generalisation seem to be highly related, but the latter is not mentioned. Aren't resilience and generalisation the same?
>
> Thank you for raising this important point. We agree that resilience and generalization are closely related concepts, but we deliberately distinguish them to highlight their different focuses and implications for algorithm design.
>
> In our work, resilience refers specifically to the capability of an agent to recover performance after encountering unexpected uncertainties. We formalize this as performance under a new state distribution $s^u$, which represents the state of the system after an external perturbation has occurred. These post-uncertainty states are often far from nominal initial states and are not under the control of the agent.
>
> In contrast, generalization in RL typically involves evaluating how well a policy performs when trained and tested on different initial states or environment dynamics. The variations are usually known or deliberately randomized (e.g., random starts or human-initialized states as discussed in [1]). The key distinction lies in the nature of the initial state distribution:
>
> In generalization, the agent starts from a known distribution of initial states, potentially with limited or no exposure to unseen variations.
>
> In resilience, the agent starts from an unknown, perturbed state $s^u$ induced by adversarial or stochastic uncertainties, often in parts of the state space rarely encountered during training.
>
> Thus, we view resilience as a special, practically motivated subset of generalization that focuses on the real-world challenges of recovery and generalization under unknown perturbation budgets. Moreover, because the perturbed state $s^u$ is the result of certain perturbations, it opens up new algorithmic possibilities that go beyond classical generalization, such as designing policies that can actively adapt or recover given a disturbed state. We will revise the manuscript to include this discussion and clarify how resilience relates to, but differs from, standard notions of generalization.
>
> [1] Revisiting the Arcade Learning Environment: Evaluation Protocols and Open Problems for General Agents, JAIR 2018.
>
> > Q4: It is counterintuitive to define resilience as agents evolving in an environment with a new initial state distribution. It makes sense formally, but it does not help the intuition behind.
>
> To reconcile the difference between formal definition and intuition, we revise resilience as starting at time t, after encountering uncertainties. Thus, the task starts at an arbitrary time t with an inferior state for recovery. The revised equation is: $J^{\text{resilience}}(\pi) = \mathbb{E}\_{u \sim \mathcal{U}} [ \mathbb{E}\_{s^u \sim \rho^u} \mathbb{E}\_{\pi} [ \sum\_{t}^\infty \gamma^t r\_t | s\_t = s^u ] ]$.
>
> > Q5: It is always difficult to choose hyperparameter and their values. What are the default choices and why you selected them?
>
> Currently, there do not exist a universal hyperparameter for MARL algorithms. In our paper, we select parameters that are commonly used in existing codebases, including the codebase of MAPPO [1] and HAPPO [2]. These codebases includes a set carefully designed hyperparameters that works well for MPE, SMAC and MAMujoco, which varies in different tasks. Among all hyperparameters in these codebase, we conduct a pilot study to select hyperparameters that works well in most of our real-world tasks, and select other hyperparameters as alternate solutions.
>
> [1] The Surprising Effectiveness of PPO in Cooperative, Multi-Agent Games. NeurIPS 2022.
>
> [2] Heterogeneous-agent reinforcement learning. JMLR 2024.
>
> > Q6: How would you define/compute the threshold in uncertainty when robustness becomes resilience?
>
> In fact, such threshold can be hard to define mathematically, but more of an application perspective. Imagine a power grid in urban areas, in common days, there may be small perturbations to the power grid, such as sensor errors or one user request more power than usual. In such cases with small perturbations, the power grid should be robust, such that it can serve all users despite these small variations.
>
> In rare events such as earthquake, the power grid may collapse physically and receives zero request if facing blackout, or high request if facing a short-circuit. In such cases, after engineers have fixed the power grid, it is crucial for the system to continue functioning well after recovery.
>
> > Q7: insights on how MADDPG and MAPPO/HAPPO behave differently facing different types of uncertainty. Maybe discrete/stochastic policy is the answer?
>
> Please allow us to explain this in detail. If using discrete or stochastic policy is the answer, MAPPO and HAPPO should be robust to action uncertainties by using stochastic policies. However, our finding shows MADDPG is more robust to action uncertainties than MAPPO/HAPPO. To explain this, we empirically find the exploratory noise of MAPPO and HAPPO converge to a small value when training is over, bringing limited explorations in action space, and more deterministic policies. In contrast, in MADDPG, the exploratory noise in action space remains constant. Since the policy is constantly exploring in action space, it becomes more robust against action uncertainties.
>
> > Q8: Alternative approaches than hyperparameter tuning, like train agents in robust and resilient configurations to investigate whether agents can learn them directly? This could also improve the claim that being robust for one type of uncertainty does not imply being robust to all, and to improve all claims anyway.
>
> We fully admit that hyperparameter tuning is not the only solution to robust and resilient MARL. The problem lies in robust MARL itself.
>
> First, training MARL against uncertainties is commonly referred as robust MARL, which is a complex process that requires high expertise. Existing robust MARL algorithms are specific to their own type of perturbation and their backbone MARL algorithms. For example, EIR-MAPPO [11] works only on MAPPO backbone with action uncertainty applied to one agent, M3DDPG [24] works for MADDPG backbone only with action uncertainty applied to all agents. Similarly, in observation uncertainty, noise can be applied to each agent [8] or all agents [9]. Moreoverr, as reported by EIR-MAPPO [11], since their method is designed against action uncertainties, they are not robust to obervation uncertainties. To the best of our knowledge, there do not exist a single work on robust MARL that (1) can be applied to any backbone, (2) works for perturbations on one, several or all agents and (3) works for perturbations over observation, environment and actions.
>
> The method of training MARL against uncertainties is most similar to the work of [1], which use curriculum learning to train robust MARL against diverse uncertainties in action, observation and rewards. However, the authors of this paper found training MARL against diverse uncertainties is hard, can often harm convergence, and can be robust against at most two uncertainties. We additionally conducted a prelinminary study ourselves, and found the same difficulty on convergence.
>
> [1] Robustness to Multi-Modal Environment Uncertainty in MARL using Curriculum Learning. NeurIPS 2023.
>
> Second, we have compared hyperparameter tuning on ERNIE (Fig 9), a general-purposed robust MARL that turns uncertainty into regularization terms, which we find to have good convergence properties empirically. While ERNIE is itself a robust MARL method, performing hyperparameter tuning further enhance its performance. Thus, we deem hyperparameter tuning and robust MARL methods are orthorgonal ways to achieve robust and resilient MARL.

---

> > ### Comment · Reviewer_PfFs · 2025-08-04
> >
> > Thanks for the thorough answers and different details.
> > Can you comment on how you will modify the paper based on the statement, or which statement of the paper covers this?
> > I believe some of your content provided here is better than some of the papers.
> >
> > While I would be happy to understand how you integrate each statement, here are some comments on three of them.
> >
> > On Q3:
> > In Machine learning, not only in RL, generalization refers to the model's ability to handle unseen data.
> > My comment focuses on this definition, rather than a narrow one that generalizes to a known initial state distribution.
> > The word generalization is not written in the paper and must be. This statement can add value.
> >
> > On Q6:
> > I disagree.
> > Especially by using your three objectives and defining bounds on the uncertainty.
> > While I agree that these bounds may be hard to quantify in practice, or would be a choice.
> >
> > On Q7:
> > I don’t think the choice of algorithm is the answer, but I firmly believe that it plays a role.
> > I am not sure about your justification, but discussions can improve the paper.
> > The noise of the stochastic policies decreases through time and is linked to hyperparameter choices.
> > Your statement is thus correct at the end of the training.
> > How is this noise higher than that of MADDPG at the beginning?
> > Don’t you think stochastic policies suffer from their initial high stochasticity?

---

> > ### Author Response · Authors · 2025-08-05
> > **Response to reviewer (part 1)**
> >
> > Dear Reviewer,
> >
> > Thanks for the feedback. We will include every point of the discussion with all reviewers in our final version. Specifically, we will:
> >
> > * Revise the Section 2: Robustness and Resilience to include the discussions on relationship between generalization and resilience, relation with robustness and resilience, discussions and case study on the relation of robustness and resilience.
> >
> > * Provide a clear and concise summarization of our findings.
> >
> > * Include additional discussions on the impact of hyperparameters.
> >
> > * Reasons for selecting current set of hyperparameters (depends on current codebase).
> >
> > * Include additional discussions, such as the different behavior of MADDPG with MAPPO/HAPPO.
> >
> > * Remove excessive reference to the codebase and move some technical details to Appendix, making the main paper more concise and enable readers to find all technical details in Appendix.
> >
> > * Other suggestions by reviewers, such as moving Related Work section to appear earlier, additional details of our environments, etc.
> >
> > * Add additional suggestions on further direction for robust MARL in conclusion for potential researchers.

---

### Official Review · Reviewer_GvL9 · 2025-06-28

**Clarity:** 3
**Significance:** 3
**Originality:** 3
**Rating:** 4
**Confidence:** 4

**Summary:**

The paper studies the robustness and resilience of policy learning for Multi-agent reinforcement learning under uncertainties. It considers three types MARL algorithms: MAPPO, HAPPO and MADDPG, and investigates the effect of typical hyper-parameters on the policy robustness and resilience. The paper evaluates different combinations of hyper-parameters across a wide range of benchmark tasks. The conclusions show that the relationship between cooperation and robustness/resilience may not be strongly corrected, and the hyper-parameter tuning is crucial for policy learning under different uncertainties.

**Questions:**

These questions do not affect my evaluation score.
1. “Use a higher learning rate (LR) for the critic.” It’s a bit unclear if this high learning rate is caused by the insufficient number of mini-epochs the critic is optimized with. Other factors could also affect this conclusion, e.g., the batch size.
2. “As for PopArt, its normalization benefits may be unnecessary when reward scales remain stable, leading to inconsistent effectiveness across different tasks.” It’s a bit surprising to see PopArt leads to inconsistent effectiveness when the reward scales remain stable. I was wondering when PopArt was used whether the advantage is also scaled according in gradient estimate.
3. the conclusion that MADDPG is not benefitting from the increased exploration is also a bit surprising to me. MADDPG has different exploration schemes than MAPPO and HAPPO. This direction comparison in terms of entropy bonus between these methods may not stand.
4. in the definition of robustness and resilience, the inner expectations are both with $u$, which refers to some uncertainty sampled from a set. However, in the resilience description “the system evolves without additional uncertainties”, not sure if there is a typo in the inner expectation for resilience.

**Ethical Concerns:**

["NO or VERY MINOR ethics concerns only"]

**Final Justification:**

my main concern remains as the robustness and resilience considered in this paper are defined within a limited scope. some other implementation-related issues seem not answered directly as the authors just referred them as the default setups by MAPPO. I think the paper is around the borderline.

**Limitations:**

yes

**Paper Formatting Concerns:**

no major formatting issues.

**Quality:**

3

**Strengths And Weaknesses:**

### Strength
**Originality**: the paper clearly defines the robustness and resilience for policy performance in multi-agent reinforcement learning. It considers both general and algorithm-specific hyperparameters and their impact on the policy robustness and resilience.
**Quality & Significance**: the evaluation is conducted on four real-world environments, with the consideration of policy transfer to physical robots and a wide range of distinct uncertainties in observations, actions, and environments. The evaluation dimensions cover the correlation among cooperation, robustness and resilience, generalization of robustness across different uncertainties, and the effective hyper-parameter tuning.

### Weakness
**Significance**: the robustness and resilience considered in this paper are both defined over a pre-defined set of uncertainties, which might limit the scope of possible application scenarios. For example, the uncertainties might switch from one type to another and might yield non-stationary property.

---

> ### Author Rebuttal · Authors · 2025-07-30
>
> We thank Reviewer GvL9 for appreciating the novelty, large-scale empirical study, and significance of our findings. A detailed response is provided below.
>
> > Q1: the robustness and resilience considered in this paper are both defined over a pre-defined set of uncertainties, which might limit the scope of possible application scenarios. For example, the uncertainties might switch from one type to another and might yield non-stationary property.
>
> Thanks for the observation. We answer this from three perspectives. First, we control the variable of noise to limit the amount of randomness faced in our analysis. The variable control is significant for our statistical tests since only one variable is changed, which ease our effort to study the impact of each noise/algorithm/tasks/hyperparameters.
>
> Second, using mixed type of uncertainties may introduce large variations in experiments. For example, if we select some type of uncertainties to appear after another uncertainty at some time, we must control the time, type and magnitude of uncertainty we applied, which makes the evaluation computationally intractable and introduce large variations during evaluations.
>
> Third, existing robust MARL literature use bounded noises to evaluate their performance. We select the current setting to connect with current literatures.
>
> We fully agree that mixing these uncertainties is important for comprehensive evaluation. We will add this as a function in our codebase to support real-world evaluation of these non-stationary noises.
>
>
> > Q2: It’s a bit unclear if higher LR of critic is caused by the insufficient number of mini-epochs the critic is optimized with. Other factors could also affect this conclusion, e.g., the batch size.
>
> Our code is based on the widely used codebase of MAPPO [1], where actors and critics are optimized with the same mini-epochs. The batch size also inherits the default of MAPPO, which is the same for actors and critics. Thus, higher LR of critic indeed helps, and suggests a potential optimization direction for MARL.
>
> [1] The Surprising Effectiveness of PPO in Cooperative, Multi-Agent Games. NeurIPS 2022.
>
>
> > Q3: It’s a bit surprising to see PopArt leads to inconsistent effectiveness when the reward scales remain stable. I was wondering when PopArt was used whether the advantage is also scaled according in gradient estimate.
>
> Our implementation again follows the official implementation of MAPPO. They use PopArt to normalize the learning process of the critic. When computing the advantage for the actor, the value prediction is first denormalized by the inverse PopArt transform and then the advantage is Z-score normalized for policy gradient.
>
>
> > Q4: MADDPG is not benefitting from the increased exploration is also a bit surprising to me. MADDPG has different exploration schemes than MAPPO and HAPPO. This direction comparison in terms of entropy bonus between these methods may not stand.
>
>
> Thanks for the insightful comment. We agree that MADDPG’s exploration scheme differs from the entropy-based exploration used in MAPPO and HAPPO, and that direct comparisons must be interpreted with care.
>
> In MAPPO/HAPPO, the entropy bonus is a soft constraint, which encourage explorration at the beginning, while the policy can still converge to a relatively deterministic policy which reach optimal cooperation.
>
> In contrast, the exploratory noise of MADDPG is additive action noise, which is enforced throughout the training process and does not decay naturally. This can lead to suboptimal behavior, especially when large noise values perturb the agent into poor or unstable regions of the state space. Empirically, we find that increasing MADDPG’s action noise did not lead to better performance. We believe this is because excessive noise degraded the quality of the agent’s behavior, making it harder to maintain meaningful strategies.
>
> We will add the analyzation of on-policy and off-policy exploration scheme above in final revised version for further analyzation.
>
> > Q5: In the resilience description “the system evolves without additional uncertainties”, not sure if there is a typo in the inner expectation for resilience.
>
> Thanks for pointing this mistake. The inner expectation should not additionally condition on u. We will correct this in final revision.

---

> ### Comment · Reviewer_GvL9 · 2025-08-05
> **Thanks for the response**
>
> Thanks for the response. I will keep my score unchanged.

---

### Official Review · Reviewer_bhkV · 2025-07-02

**Clarity:** 3
**Significance:** 3
**Originality:** 2
**Rating:** 4
**Confidence:** 2

**Summary:**

This paper presents a large-scale and comprehensive empirical study on robustness and resilience in cooperative MARL. The authors clearly distinguish and formalize robustness (the ability to withstand uncertainties) and resilience (the ability to recover from severe disruptions). Through 82,620 experiments across four real-world-inspired environments, the study systematically evaluates three widely used continuous-control MARL algorithms (MAPPO, MADDPG, HAPPO) under 13 types of uncertainty and 15 hyperparameters. The key findings are: (1) while optimizing for cooperation can improve robustness and resilience under mild uncertainty, this correlation weakens as perturbations intensify; (2) robustness and resilience do not generalize across uncertainty modalities or agent scopes; and (3) hyperparameter tuning can have a greater impact on these properties than the choice of MARL algorithm, with some standard practices (such as parameter sharing, GAE, and PopArt) sometimes harming robustness, while strategies like early stopping, higher critic learning rates, and Leaky ReLU are consistently beneficial.

**Questions:**

1. To what extent do the main findings generalize to other MARL algorithms, such as value decomposition or graph-based methods? Have the authors considered extending their analysis to these additional families?
2. Many results are shown across four environments, but how confident are the authors that key observations would hold in other domains, such as additional robotics, control, or RL tasks (e.g., gym benchmarks)?
3. Have similar sensitivities to hyperparameter tuning been observed in more recent robust MARL methods (e.g., distributional RL, opponent modeling, etc.)? Would direct empirical comparison with these advanced baselines alter the main conclusions?
4. Could the authors provide further justification for the chosen hyperparameter ranges, and clarify whether the reported benefits and negative effects persist under different normalization or reward structures?
5. Have the authors considered whether noise effects accumulate in later-collected samples as training progresses? Is there any empirical evidence or analysis regarding this potential issue?
6. The study primarily uses 5 random seeds per experiment. Would increasing this number improve the statistical confidence of the results, especially for small effect sizes?
7. Given the experimental scale, what are the computational requirements for reproducing the main findings? Are all tested environments and uncertainty settings easily accessible and reproducible via the released code?
8. There are repeated references to the code link and several instances of redundant information. Would the authors consider streamlining these sections for clarity and conciseness?

**Ethical Concerns:**

["NO or VERY MINOR ethics concerns only"]

**Final Justification:**

The authors do not propose any new algorithms or theories, which is my main hesitation. However, the empirical study is undeniable. Therefore, the final decision depends on whether the empirical study alone is sufficient.

**Limitations:**

Yes

**Quality:**

3

**Strengths And Weaknesses:**

**Strengths**: The greatest strength of this paper lies in the scale, rigor, and comprehensiveness of its empirical investigation—spanning over 82,000 experiments across four real-world-inspired environments, 13 types of uncertainty, and 15 hyperparameters. This enables robust, statistically meaningful conclusions and demonstrates a level of methodological thoroughness rarely seen in MARL research. The authors provide clear and rigorous distinctions between robustness and resilience, offering a practical framework and formal metrics that fill an important gap in the literature. Their systematic evaluation covers multiple algorithms and environments, with a modular codebase that enhances reproducibility and extensibility for practitioners. The study’s concrete findings, such as the outsized impact of hyperparameter tuning (often exceeding that of algorithmic choice), and the counterintuitive result that standard practices like parameter sharing, GAE, and PopArt can be detrimental under uncertainty, offer actionable guidance for the community.

**Weaknesses**
1. The paper does not introduce new algorithms, training paradigms, or theoretical guarantees. Its contributions primarily lie in systematic evaluation and formalization, which limits the originality and theoretical depth.
2. The study centers on three mainstream MARL algorithms (MADDPG, MAPPO, HAPPO), and while these are representative, the results may not generalize to alternative MARL paradigms (e.g., value decomposition, graph-based, or distributional RL methods).
3. Some conclusions may be specific to the chosen environments or implementation details, such as the negative effects of parameter sharing, GAE, and PopArt, or the universality of certain tuning “tricks”.
4. There is limited presentation of raw numerical results, confidence intervals, or effect sizes in the main text; detailed per-task outcomes are relegated to the appendix, reducing transparency about the practical impact on individual tasks.
5. The empirical comparison does not include more recent robust MARL methods (e.g., distributional RL, adversarial or opponent modeling).
6. The large breadth of experiments and the dense presentation may overwhelm readers, and some methodological choices (such as rationale for selected hyperparameter ranges or interaction effects) could be discussed more clearly for accessibility.

---

> ### Author Rebuttal · Authors · 2025-07-30
>
> We thank Reviewer bhkV for appreciating the scale, rigor, comprehensiveness, codebase contribution, statistical test and actionable guidance of our paper. A detailed response is provided below.
>
> > Q1: The paper do not introduce new algorithms and new theoretical guarantees.
>
> Thank you for the comment. It is true that our paper does not propose a new algorithm or theoretical guarantee. However, the core aim of this work is to provide a comprehensive empirical study on robustness and resilience in MARL, which, to our knowledge, is still under-explored.
>
> Rather than introducing a single new algorithm, we contribute a set of key empirical findings and actionable insights that can guide the development of an entire class of more robust and resilient MARL algorithms. We believe this kind of contribution can be equally, if not more, valuable to the community, as it identifies where and how current methods fail and what design principles matter most under different uncertainty conditions.
>
> For example, in Section 4.3, we give suggestions on hyperparameters that improves cooperation, robustness and resilience together. In Section 4.4, we show applying these hyperparameters can improve MARL performance largely on both purely cooperative, and robust MARL. Similarly, in Section 4.2, we should robust MARL should be evaluated across uncertainty modalities and agent scopes, which highlight limitations of current evaluation and point to new algorithmic opportunities.
>
> > Q2: The results may not generalize to alternative MARL paradigms (e.g., value decomposition, graph-based, or distributional RL methods)
>
> Thank you for the valuable feedback. While our focus is on policy-based MARL, many of our findings remain relevant to alternate MARL paradigms, and we identify this as a promising direction for future work.
>
> First, our real-world environments requires continuous control without graph structure. Thus, mainstream value decomposition based method like QMIX are not applicable and the graph sturcture is not available. We acknowledge the importance of these works and will discuss them in the related work.
>
> Second, while our study does not directly evaluate these methods, many of these paradigms are built on policy gradient frameworks. Thus, many insights should still transfer, such as the benefits of higher critic learning rates, parameter sharing, GAE, PopArt, and exploration strategies.
>
> Third, value decomposition methods assume all agents contribute positively, which fails in adversarial settings with compromised agents. Graph-based MARL may be more vulnerable to critical agents due to inter-agent connectivity, while distributional RL may offer greater robustness via stochasticity. However, evaluating these would require ~80,000 additional experiments, which is beyond our current scope.
>
> Finally, our codebase (Appendix A.4) support new algorithms that meets our interface requirement and can be used to evaluate alternative MARL paradigms under adversarial perturbations. Our codebase support QMIX which is based on value decomposition currently. We will include this as a limitation and an invitation for follow-up work in the final version.
>
> > Q3: Some conclusions may be specific to the chosen environments or implementation details.
>
> We believe our findings are general enough for the following reasons. While no benchmark can fully represent MARL, our environments span over diverse and representative tasks of MARL.
>
> | Environment | Task Type   | Control Type  | Episode Length | Engine/Source | Data Source  | Goal  | Challenge |
> |------|-----|-------|------|------|-------|---|----|
> | DexHand | Multi-robot Manipulation  | Continuous| ~80  | Isaac Gym | Real-world Robots | Hand Manipulation  |  Precise Control   |
> | Quads | Multi-robot navigation  | Continuous | ~1600| OpenAI Gym | Real-world Robots | Reach Target & avoid collision |  Long-range multi-robot task assignment|
> | Traffic| Network control   | Discrete| ~1000| SUMO  | Real-world traffic | Optimize throughput  | Long-range control of multiple agents   |
> | Voltage| Network control   | Continuous | ~200| IEEE Standard | Real-world power grid  | Voltage Management |  Complex and noisy dynamics|
>
> Under diverse environments and uncertainties, our findings suggests certain tuning directions that are consistent and statistically significant across all settings, including higher critic LR, early stopping and Leaky ReLU. Moreover, while parameter sharing, GAE and PopArt are treat as default by previous researchers, our finding shows it is merely a design choice, revealing blind spots that researchers may overlook. While these findings may not help for any task, algorithm and random seed, they are a promising direction for optimization and helps at high probabilities.
>
> > Q4: There is limited presentation of raw numerical results, confidence intervals, or effect sizes in the main text. Per-task outcomes are relegated to the appendix.
>
> We will share the raw reward score under each configurations and uncertainties after acceptance of our paper and include additional raw numerical results in Appendix for further analysis. Since we are sharing insights of our evaluation, the raw records may be messy and requires time and effort to understand.
>
> As for confidence intervals and effect sizes, we will add them in main paper in final revision.
>
> > Q5: The empirical comparison does not include more recent robust MARL methods (e.g., distributional RL, adversarial or opponent modeling).
>
> First, please allow us to clarify we have included comparison with ERNIE in Fig 9, a general-purposed robust MARL method. The result shows our selcted hyperparameters also works for ERNIE. Given the fact that many robust MARL algorithms are based on MARL backbones like MAPPO, our result helps these robust MARL methods as well.
>
> Second, robust MARL is complex. Most algorithms are specific to their own type of perturbation and their backbone MARL algorithms. For example, EIR-MAPPO [11] works only on MAPPO backbone with action uncertainty applied to one agent, M3DDPG [24] works for MADDPG backbone only with action uncertainty applied to all agents. Similarly, in observation uncertainty, noise can be applied to each agent [8] or all agents [9]. Moreoverr, as reported by EIR-MAPPO [11], since their method is designed against action uncertainties, they are not robust to obervation uncertainties. To the best of our knowledge, there do not exist a single work on robust MARL that (1) can be applied to any backbone, (2) works for perturbations on one, several or all agents and (3) works for perturbations over observation, environment and actions. As such, we compare ERNIE, which is the robust MARL algorithm that have the highest potential to deal with all types of uncertainties.
>
> > Q6: The large breadth of experiments and the dense presentation may overwhelm readers, and some methodological choices (such as rationale for selected hyperparameter ranges or interaction effects) could be discussed more clearly for accessibility.
>
> We will include a takeaway table for readers to summarize our main points below:
>
> |Scenario|Suggestion|Ref.|
> |---|------|-----|
> |Small uncertainties|Optimize cooperation|§4.1|
> |Large uncertainties|Use robustness/resilience methods|§4.1|
> |Obs. uncertainties|MAPPO/HAPPO best|§4.1|
> |Action uncertainties|MADDPG best|§4.1|
> |Modalities|Evaluate obs/env/action separately|§4.2|
> |Agent scopes|Test individual/global perturbations|§4.2|
> |Early stop|On|§4.3|
> |Critic LR|> Actor LR|§4.3|
> |Activation|Leaky ReLU|§4.3|
> |GAE|Off|§4.3|
> |PopArt|Off|§4.3|
> |Param sharing|On (homogeneous), Off (heterogeneous)|§4.3|
> |Exploration|High (MAPPO/HAPPO), Moderate (MADDPG)|§4.3|
>
> We select hyperparameters that are commonly used in existing codebases, including the codebase of MAPPO [1] and HAPPO [2]. These codebases includes a set carefully designed hyperparameters that works well for MPE, SMAC and MAMujoco, which varies in different tasks. Among all hyperparameters in these codebase, we conduct a pilot study to select hyperparameters that works well in most of our real-world tasks, and select other hyperparameters as alternate solutions.
>
> [1] The Surprising Effectiveness of PPO in Cooperative, Multi-Agent Games. NeurIPS 2022.
>
> [2] Heterogeneous-agent reinforcement learning. JMLR 2024.
>
> For interaction effects, we conduct this anaysis to check if the hyperparameters are still effective when combining with each other. we will move some details in Appendix B.7 to main paper for increased clarity.
>
> > Q7: Have the authors considered whether noise effects accumulate in later-collected samples as training progresses?
>
> The training progress is done in ideal simulation environment, mimicing current MARL training paradigm so we do not add noise in sample collection. Adding noise during training is often framed as backdoor or poisoning attacks, which follow a different theoretical paradigm and is out of the scope of our paper.
>
> > Q8: Would increasing random seeds affect statistical confidence?
>
> Please allow us to clarify that all our results are averaged across multiple runs, tasks and algorithms. Thus, while we tested for 5 random seeds, the sample sizes are more than 100 in Section 4.2 and 4.3. Thus, while we admit more random seeds are always better, we believe the number of samples are sufficient to make statistically significant claims.
>
> > Q9: What are the computational requirements for reproducing the main findings?
>
> Our full experiment takes ~230K GPU hours, measured in GTX 4090. For better reproducability, we will share the raw reward score under each configurations for further analysis.
>
> > Q10: repeated references to the code link and several instances of redundant information.
>
> Thanks for the suggestion. We will reduce these redundant information in our final revision.

---

> > ### Comment · Reviewer_bhkV · 2025-08-04
> >
> > Thanks for the rebuttal. I will keep my score for potential acceptance.

---

### Official Review · Reviewer_W4zq · 2025-07-03

**Clarity:** 3
**Significance:** 2
**Originality:** 2
**Rating:** 4
**Confidence:** 4

**Summary:**

This paper explores the importance of tuning hyper parameters in MARL systems with the goal of understanding their roles in cooperation, robustness and resilience.

It makes the clear distinction between that robustness is ability to deal with constant perturbations while resilience is the ability to recover from disruptions.

They test a combination of 82,620 different experiments. Varying hyper parameters, model designs and environments.
The statistical analysis identifies a few compelling ideas and correlations.

1.	Optimizing cooperation under mild uncertainty enhances both resilience and robustness, though this relationship diminishes as uncertainty grows.

2.	The modality of the uncertainty affects different policies in different ways. For example, MAPPO and HAPPO are more robust to observation uncertainties and MADDPG is better with action uncertainty. (This posses the question of whether there is a model that can incorporate the strengths of both of these.)

3.	Parameter sharing appears to reduce cooperation, robustness, and resilience, except for improvements in SMAC.
An interesting question to ask is why does parameter sharing help in SMAC but hinder performance in MPE and Multi-Agent Mujoco? Is this because the agent  roles and objectives are the same for some agents in SMAC but not in the others? (example, zealots have the same role and objective and will likely benefit from the same parameters, while a particle in MPE has a different objective to another particle and thus benefits from a different policy entirely.)

4.	GAE, and PopArt can hurt robustness, as they found in MPE and SMAC, but helps in Multi-agent Mujoco. They suggest that it’s because of the dense and stable reward structure. Does this not just mean it is a design choice we should be tuning if we know about the reward structure?

5.	They find that early stopping helps. Suggesting saving a model at a point where it is optimal over cooperation, robustness and resilience is better than saving the final model in many cases. This begs the age old question, at what point should we be stopping?

6.	High critic learning rates (higher than actors) and using Leaky ReLU help with robustness. The critic updating faster than the actors is suggested to have more robust results, and Leaky ReLU is suggested to solve the dead neuron problem that arises with ReLU. Are there environments that this might not be the case?

**Questions:**

1. Is there possibly a more principled approach to the hyper parameter search than just checking over a range? This feels like it would depend heavily on the environment and the model being checked.

2. Are you sure your general statements about GAE, PopArt, etc... can be generalized? Especially considering the dense reward in Mujoco seemingly benefiting from it.

**Ethical Concerns:**

["NO or VERY MINOR ethics concerns only"]

**Final Justification:**

I will keep my score.
I lean toward acceptance based on the share magnitude of the experiments run, and I believe there are good insights to be drawn from these experiments, with the code base also allowing for ease of implementation for future researchers.

**Limitations:**

Yes.

**Paper Formatting Concerns:**

Should the related work not be earlier in the paper?

**Quality:**

2

**Strengths And Weaknesses:**

Strengths:

1.	The comprehensive statistical analysis on the different experiments are thorough and do highlight factors we should consider.

2.	The work covers many different hyper parameters that can be chosen and shows reasons why one size fits all can’t be the solution for MARL (However I am not convinced that the correct conclusion is always drawn. The authors even find exceptions to their claims, eg parameter sharing being bad except in SMAC).

3.	There is a code base that can help anyone tune their models. Which is useful since most studies don’t focus enough on the hyper parameter tuning.

4.	The paper correctly distinguishes between robustness and resilience, a concept that many MARL works lack clarity on.

5.	The insight that GAE and PopArt are probably only useful in high density reward and harmful at other times is useful.

Weaknesses:

1.	The paper generalizes claims about concepts, while directly giving a contradictory example. See the example with parameter sharing and SMAC, or GAE and PopArt with Mujoco. I believe there is likely a more principled manner to make these claims, since the exceptions aren't always explained.

2.	The hyper parameter search appears to just be a sweep over a certain range, is there not a more principled manner to do this?

3.	Minor: Typically, the related work is stated at the start of the paper to help frame the work.

---

> ### Author Rebuttal · Authors · 2025-07-30
>
> We thank Reviewer W4zq for appreciating the novelty, large-scale empirical study, codebase contribution, statistical contributions and findings of our paper. A detailed response is provided below.
>
> > Q1: The paper generalizes claims about concepts, while directly giving a contradictory example. Likely a more principled manner to make these claims, since the exceptions aren't always explained.
>
> 1. When and why parameter sharing do not work?
>
> We believe parameter sharing helps in environments where agents are more homogeneous like SMAC, where agents are of the same type (eg, zealot, marine). However, it do harm performance for Voltage, Traffic, MPE and MAMujoco since in these environments, agents are heterogeneous (Line 307-311). For Dexhand and Quads, the effect is mixed and non-significant. While we do believe agents benefit from parameter sharing when they are homogeneous, there do not exist a quantitative way to measure how "homogeneous" the environment is since it is highly task-specific. A safe conclusion is to treat parameter sharing as a design choice, rather than default implementation, especially in environments with heterogeneous agents.
>
> Our claim is reinforced in HATRPO [1], where they show parameter sharing may result in suboptimal actions since it pose an important constraint that the parameters of all agents are identical. The Proposition 1 made by HATRPO even show parameter sharing brings exponentially worse outcome by constructing a negative example. On MAMujoco environment, they also find parameter sharing "introduces extra variance to training, harms the monotonic improvement property, make algorithm converge to suboptimal policies, and the problem gets more severe in the task with more agents".
>
> [1] TRUST REGION POLICY OPTIMISATION IN MULTI-AGENT REINFORCEMENT LEARNING. ICLR 2022.
>
> 2. GAE/Popart. is it just a design choice if we know the reward structure?
>
> While we do find GAE and PopArt helps in MAMujoco with dense and stable reward structure, and can have negative impact in our 4 real world environment, SMAC and MPE with statistical significance, we shall additionally remind readers that such claims are made empirically and statistically, which will not hold for every case. However, the statistical significant result do suggest a potential direction for tuning, which have high probability (ie, statistically significant) to yield positive results.
>
> 3. While early stopping helps, at what point we should stop?
>
> If we have a perfect measure of model performance, we should stop exactly at that point. However, in MARL community, a common way is to optimize cooperative performance greedily, which we find it will hurt robustness and resilience after a certain point due to overfitting. As such, while our focus on cooperation, robustness and resilience may not be a perfect measure, it can be a good enough starting point to start with in real world, and is much better than optimizing cooperation performance only. In practice, engineers can estimate the range of uncertainties in their own tasks, and perform early stop accordingly.
>
> 4. While high critic learning rates and leaky relu works better, are there any environment not the case?
>
> In fact, there are environments and uncertainties that are not the case. For example, higher critic LR slightly hurts resilience in Dexhand and Leaky ReLU slightly hurts resilience for MADDPG. However, in all experiments we conduct, higher critic LR works better in 153/161 cases, and leaky relu works better in 148/161 cases. As a consequence, trying it first is likely to return positive result on average.
>
>
> > Q2: The hyperparameter search appears to just be a sweep over a certain range, is there not a more principled manner to do this? it will heavily depend on the environment and models.
>
> To the best of our knowledge, there do not exist a theoretically rigorous way to define ideal hyperparameter sweeps (otherwise we can tune MARL optimally and analytically!). Thus, we select parameters that are commonly used in existing codebases, including the codebase of MAPPO [1] and HAPPO [2]. These codebases includes a set carefully designed hyperparameters that works well for MPE, SMAC and MAMujoco, which varies in different tasks. Among all hyperparameters in these codebase, we conduct a pilot study to select hyperparameters that works well in most of our real-world tasks, and select other hyperparameters as alternate solutions.
>
> [1] The Surprising Effectiveness of PPO in Cooperative, Multi-Agent Games. NeurIPS 2022.
>
> [2] Heterogeneous-agent reinforcement learning. JMLR 2024.
>
> > Q3: the related work is stated at the start of the paper to help frame the work.
>
> Thanks for the suggestion. We will move that section in final version.
>
> > Q4: Can findings on general statements be generalized?
>
> We believe our findings can be generalized as follows. Our findings suggests certain tuning directions that are statistically significant (i.e., likely to work on average), including higher critic LR, early stopping and Leaky ReLU. Moreover, while parameter sharing, GAE and PopArt are previously regarded as panacea to cooperative MARL, previous researchers may treat this as default. Our finding shows it is merely a design choice, revealing blind spots that previous researchers may overlook. We do not claim our findings to be a silver bullet that helps any task and any algorithm under any random seed, but it is a promising direction for optimization.
>
> As for the claims in Section 4.1 and 4.2, we believe those evidence are general enough to guide the design of robust MARL algorithms. In Section 4.1, we suggest focusing on the more harmful attacks. In Section 4.2, we suggest focusing on attacks over all modality and agent scopes. We summarize our general suggestions as follows:
>
>
> | Scenario / Task                          | Suggested Practice                                         | Reference     |
> |------------------------------------------------|----------------------------------------------------------------|-------------------|
> | Small uncertainties                            | Optimizing for cooperation is sufficient                       | Section 4.1       |
> | Large uncertainties                            | Requires dedicated robustness or resilience strategies         | Section 4.1       |
> | Robustness to observation uncertainties        | MAPPO / HAPPO perform best                                     | Section 4.1       |
> | Robustness to action uncertainties             | MADDPG performs best                                           | Section 4.1       |
> | Uncertainty across modalities                  | Evaluate observation, environment, and action perturbations separately | Section 4.2       |
> | Uncertainty across agent scopes                | Test both individual-agent and global-agent perturbations      | Section 4.2       |
> | Early stopping                                 | Should always be enabled                                       | Section 4.3       |
> | Critic learning rate                           | Should be higher than actor learning rate                      | Section 4.3       |
> | Activation function                            | Use Leaky ReLU                                                 | Section 4.3       |
> | GAE (Generalized Advantage Estimation)         | Disable                                                        | Section 4.3       |
> | PopArt                                         | Disable                                                        | Section 4.3       |
> | Parameter sharing                              | Use for homogeneous agents; disable for heterogeneous agents   | Section 4.3       |
> | Exploration strategy                           | Use high entropy bonus for MAPPO/HAPPO; moderate noise for MADDPG    | Section 4.3       |

---

> > ### Comment · Reviewer_W4zq · 2025-08-05
> >
> > Thank you for your rebuttal.
> > I will keep my score.

---

### Note · Authors · 2025-08-12

Dear Reviewers and AC,

In our paper, we conduct a large-scale empirical study with over 82,620 experiments on the robustness and resilience of MARL, which includes 4 real-world environments, 13 uncertainty types, and 15 hyperparameters. Our empirical findings include suggestions for increasing the robustness and resilience of MARL including: (1) optimization direction for different magnitude of uncertainties, (2) MARL algorithm suggestion under different type of uncertainties. (3) suggestions on countering uncertainties across different modalities (observation, action, environment) and agent scopes (single/all agents) for robust MARL researchers. (4) suggestions on certain hyperparameters (Early stop, high critic LR, no param sharing, no GAE/PopArt) for better robustness and resilience for MARL practitioners.

We thank all reviewers for recommending acceptance of our paper, and their constructive feedback and discussions. Specifically, we thank Reviewer bhkV, GvL9, PfFs, W4zq for appreciating the large-scale study we conducted, significance of our findings. We thank Reviewer GvL9, W4zq for appreciating our novelty, Reviewer bhkV, W4zq for appreciating the codebase contribution. We will add all details in author-reviewer discussion to further strengthen our paper. Below are our feedback to reviewers:

* Reviewer W4zq: Thanks for suggesting acceptance of our paper. We will revise our paper to include all discussions and add a table to summarize all our suggestions for readers to get our point in one minute.

* Reviewer bhkV: Thanks for suggesting acceptance of our paper. We will revise our paper to include additional experiment details, discussions and add a table to summarize all our suggestions for readers to get our point in one minute.

* Reviewer GvL9: Thanks for suggesting acceptance of our paper. We will revise our paper to include all discussions, which further enhance our paper.

* Reviewer PfFs: Thanks for suggesting acceptance of our paper. We will revise our paper to include all discussions, especially those on the relationship between resilience and generalization, and the difference between MADDPG and MAPPO/HAPPO on robustness, which further enhance our paper.

---

### Decision · Program_Chairs · 2025-09-17

**Decision:**

Accept (poster)

**Comment:**

In this submission, the authors present a large-scale empirical study to evaluate cooperation, robustness, and resilience in multi-agent reinforcement learning (MARL) methods. The authors use 4 environments to perform a large-scale empirical study comprising over 82,620 experiments whereby they vary certain aspects of the experiments, such as the hyper parameters and model designs. The analysis of these experiments demonstrates that the relationship between multi-agent cooperation and robustness/resilience might not be strongly corrected, and that the hyper-parameter tuning remains key.

The reviewers highlighted several strengths of the paper. This includes the comprehensive and rigorous empirical evaluations and statistical analysis on a wide range of experiments. They also note that the manuscript provides a clear and practical distinctions between robustness and resilience that offers a practical formal metrics for future work. All reviewers agree that the paper is well-written and easy to follow. During the rebuttal, the authors addressed many of the reviewers concerns. While some reservations remain around limited methodological novelty, I believe that such a detailed and rigorous study would be highly appreciated by the multi-agent RL community at NeurIPS. I therefore recommend it for acceptance. I’d suggest that the authors use the reviewers' comments to improve the manuscript for the camera-ready submission.